# Co-occurring expression and methylation QTLs allow detection of common causal variants and shared biological mechanisms

Brandon L. Pierce [1,2,3], Lin Tong[1], Maria Argos[4], Kathryn Demanelis[1], Farzana Jasmine[1], Muhammad Rakibuz-Zaman[5], Golam Sarwar[5], Md. Tariqul Islam[5], Hasan Shahriar[5], Tariqul Islam[5], Mahfuzar Rahman[5,6], Md. Yunus[7], Muhammad G. Kibriya[1], Lin S. Chen[1] & Habibul Ahsan[1,2,3,8]

Inherited genetic variation affects local gene expression and DNA methylation in humans. Most expression quantitative trait loci (*cis*-eQTLs) occur at the same genomic location as a methylation QTL (*cis*-meQTL), suggesting a common causal variant and shared mechanism. Using DNA and RNA from peripheral blood of Bangladeshi individuals, here we use co-localization methods to identify eQTL-meQTL pairs likely to share a causal variant. We use partial correlation and mediation analyses to identify >400 of these pairs showing evidence of a causal relationship between expression and methylation (i.e., shared mechanism) with many additional pairs we are underpowered to detect. These co-localized pairs are enriched for SNPs showing opposite associations with expression and methylation, although many SNPs affect multiple CpGs in opposite directions. This work demonstrates the pervasiveness of co-regulated expression and methylation in the human genome. Applying this approach to other types of molecular QTLs can enhance our understanding of regulatory mechanisms.

[1] Department of Public Health Sciences, The University of Chicago, Chicago, IL 60637, USA. [2] Department of Human Genetics, The University of Chicago, Chicago, IL 60637, USA. [3] Comprehensive Cancer Center, The University of Chicago, Chicago, IL 60637, USA. [4] Division of Epidemiology and Biostatistics, University of Illinois at Chicago, Chicago, IL 60612, USA. [5] UChicago Research Bangladesh Mohakhali, Dhaka 1230, Bangladesh. [6] Research and Evaluation Division BRAC Dhaka 1212, Bangladesh. [7] International Centre for Diarrhoeal Disease Research Bangladesh, Dhaka 1000, Bangladesh. [8] Department of Medicine, The University of Chicago, Chicago, IL 60637, USA. Correspondence and requests for materials should be addressed to B.L.P. (email: brandonpierce@uchicago.edu) or to H.A. (email: habib@uchicago.edu)

Genetic variation has a substantial impact on mRNA abundance in humans[1]. Genome-wide scans to identify regions that harbor such variants, regions known as expression quantitative trait loci (eQTLs), have been conducted using RNA from a wide array of human tissue types and cell types[2], and eQTLs have been identified for the vast majority of human genes.

In addition to studies of transcript abundance, recent work has described the effects of genetic variation on other genomic and cellular phenotypes, such as DNA methylation[3–6], DNase hypersensitivity[7], histone modifications and nucleosome positioning[8], RNA splicing[9, 10], translational efficiency/ribosome occupancy[11, 12], and protein abundance[13, 14]. Because many QTLs appear to influence multiple local molecular phenotypes and since functional relationships exist between different molecular phenotypes, there is great interest in identifying variants that have coordinated effects on multiple phenotypes and understanding the mechanisms by which such variants act.

Recently, several groups have identified single-nucleotide polymorphisms (SNPs) associated with both expression of nearby genes and methylation of nearby CpG sites[15–18]. For these cis-eQTLs that also appear to be local methylation-QTL (cis-meQTLs), it is possible that co-occurring eQTLs and meQTLs share a common causal variant (CCV), suggesting a shared biological mechanism by which the causal variant influences both expression and methylation. Methylation could be reactive to expression (i.e., methylation responds to genetically determined variation in gene expression, perhaps due to a SNP's effect on transcription factor (TF) binding), or methylation could mediate the effect of the SNP on expression (i.e., increased promoter methylation suppresses TF binding). For such co-occurring eQTLs and meQTLs, several groups have developed and applied approaches intended to determine if a causal relationship exists between the local DNA methylation and expression, including likelihood-based approaches[18], Bayesian network approaches[17], and partial correlation approaches[19].

One limitation of the prior work on this topic is a lack of rigorous assessment of the hypothesis that co-occurring eQTLs and meQTLs share a CCV. Recently developed tests for "co-localization" allow one to assess whether two association signals are consistent with a shared causal variant[20]. Using summary statistics for an eQTL and a meQTL, one can estimate the probability that the eQTL and meQTL share a CCV. This information can be used to guide subsequent studies of co-occurring eQTL-meQTL pairs.

In this work, we use genome-wide data on SNPs and array-based expression and DNA methylation from peripheral blood of South Asian individuals to identify cis-eQTLs and cis-meQTLs. We describe the extent to which the observed cis-eQTLs and cis-meQTLs share CCVs using co-localization methods[20]. Using eQTL-meQTL pairs with a high probability of sharing a causal variant, we then assess the evidence that expression and methylation are causally related to one another using partial correlation analysis and mediation analysis.

## Results

**Overview**. A simple overview of our workflow for identifying eQTLs and meQTLs that are likely to share a CCV is shown in Fig. 1. A more detailed workflow is provided in Supplementary Fig. 1.

**Observed cis-eQTLs and cis-meQTLs**. We conducted genome-wide eQTL and meQTL analyses using data on 992 and 337 non-overlapping individuals from the Bangladesh Vitamin E and Selenium Trial (BEST) study (see Methods), respectively. Patterns

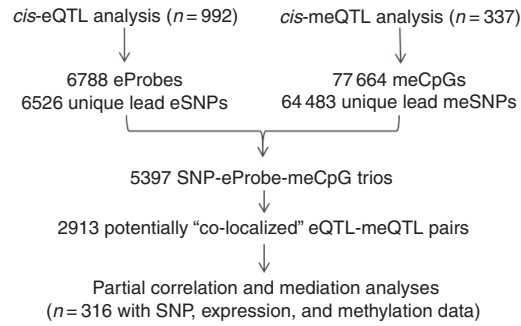

**Fig. 1** Summary of the workflow for cis-QTL, co-localization, partial correlation, and mediation analyses

**Table 1 Summary of cis-eQTL and cis-meQTL signals identified in genome-wide analyses**

|  | Cis-eQTL analysis (n = 992) | Cis-meQTL analysis (n = 337) |
|---|---|---|
| Genome-wide SNPs (n = 8 639 940) |  |  |
| Tests conducted | 52 278 603 | 994 862 964 |
| Significant SNP-probe pairs[a] | 6788 | 77 664 |
| Unique lead SNPs[b] | 6526 | 64 483 |
| Significant eQTL/meQTL genes | 5632 | 13 325[c] |

Analyses were conducted using 22 793 expression probes, 423 604 methylation probes, and 8 639 940 genotyped and imputed SNPs. SNP and probes separated by <500 kb were tested for association

Beta distribution-adjusted empirical P-values from FastQTL were used to calculate q-values and a false-discovery rate (FDR) threshold of 0.01 was applied to identify eProbes/meCpGs with a significant eQTL/meQTL

[a] Counts include SNPs in high LD

[b] For the 6526 unique lead eSNPs, after pruning out one SNP of each SNP pair with linkage equilibrium $r^2 > 0.3$, 5022 independent SNPs remain. For the unique lead 64 483 meSNPs, 29 472 independent SNPs remain by using the same pruning method

[c] n = 55 526 CpGs were assigned to a gene in Illumina annotation file

of peripheral blood cis-eQTLs and cis-meQTLs are reported in Table 1. At an false-discovery rate (FDR) of 0.01, we detected a cis-eQTL for 6788 expression probes, corresponding to 5632 genes (i.e., cis-eGenes), 6526 unique lead eSNPs, and 5022 independent eSNPs (after linkage disequilibrium (LD) pruning). At an FDR of 0.01, we detected evidence of a meQTL for 77 664 CpG sites, corresponding to 64 483 unique lead meSNPs and 29 472 independent meSNPs (after LD pruning). In all, 55 526 of these CpG sites were assigned to a total of 13 325 genes (based on Illumina's annotation). On average, lead meSNPs were ~24 kb closer to their target CpGs than lead eSNPs were to their target transcription start site ($P = 10^{-30}$, controlling for QTL P-value; Supplementary Fig. 2).

**Co-localization of cis-eQTLs and cis-meQTLs**. A total of 5192 of our 6526 unique eSNPs were associated with methylation for at least one CpG among the 77 664 CpGs with a significant meQTL (FDR of 0.01), suggesting that a substantial number of causal eSNPs may also be causal meSNPs. This overlap corresponds to 5397 unique SNP-eProbe-CpG combinations potentially representing a CCV. Using these pairs of eProbes and CpGs associated with a common SNP, we conducted a Bayesian test of co-localization[20] for each of the 5397 pairs (see Methods) in order to estimate the probability that the two association signals are consistent with a CCV (Supplementary Table 1). Bayesian co-localization analysis[20] requires specifying a prior probability for a

SNP being associated with expression only ($p_1$), methylation only ($p_2$), and both traits ($p_{12}$). Our selection of $p_1$ and $p_2$ is based on the number of eQTLs and meQTLs observed in our data (see Methods). We varied the value of $p_{12}$ ($4.4 \times 10^{-4}$, $2.9 \times 10^{-4}$, $1.45 \times 10^{-4}$, $5.8 \times 10^{-5}$, and $2.9 \times 10^{-5}$) to correspond to probabilities of a causal eSNP being a causal meSNP of 75%, 50%, 25%, 10%, and 5%, respectively, which we view as a large and reasonable range for this prior.

We designated eQTL-meQTL pairs we as probability of CCV > 80% as potentially "co-localized" pairs for further analysis (Supplementary Data 1). However, the number of pairs passing this threshold depended strongly on the value of the prior $p_{12}$,

ranging from 2913 such pairs when $p_{12}$ was set of $4.4 \times 10^{-4}$ to 266 pairs when $p_{12}$ was set to $2.9 \times 10^{-5}$ (Table 2). Due to uncertainty regarding the appropriate value for this prior, we conduct downstream analyses of "co-localized" pairs for each of the five values used for $p_{12}$.

One approach for evaluating the selection of co-localization priors is the "internal empirical calibration" approach described by Guo et al.[21] (see Methods), in which one selects the $p_{12}$ value for which the posterior expectation of co-localization is most similar to the prior expectation. This approach suggested that $4.4 \times 10^{-4}$ was the best choice for $p_{12}$ (Supplementary Fig. 3) corresponding to a probability of 75% that a causal eSNP is also a causal meSNP.

For each value of $p_{12}$, the probability of CCV was strongly related to the LD between the lead eSNP and the lead meSNP, with low LD corresponding to low probability of CCV (Fig. 2). The bimodal distribution of the CCV probabilities (for $p_{12}$ of $4.4 \times 10^{-4}$ and $2.9 \times 10^{-4}$) suggests the existence to two major types of pairs: those very likely to share a CCV and those that do not share a CCV. Six of our strongest co-localized signals (based on probability of a CCV) are shown in Fig. 3. These eQTLs and meQTLs signals are also shown, color-coded by LD with the lead meSNP and eSNP, respectively, in Supplementary Fig. 4.

**Replication of co-localization.** We obtained meQTL results from an independent set of 347 unrelated Bangladesh individuals from the Health Effects of Arsenic Longitudinal Study (HEALS) cohort (see Methods) with DNA methylation data on ~850 000 CpG sites generated using the Illumina EPIC array. Using these *cis*-meQTL results and the eQTL results described above, we were able to attempt replication for 4875 of the 5397 eQTL-meQTL

**Table 2 Number of eQTL-meQTL pairs with probability of a common causal variant (P(CCV)) > 80%, indicating likely "co-localization", for various values of the prior $p_{12}$**

| Prior probability that an eSNP is an meSNP | Prior ($p_{12}$) | Number of eQTL-meQTL pairs with P(CCV) > 80% | Proportion[a] of pairs with P(CCV) > 80% |
|---|---|---|---|
| 75% | $4.4 \times 10^{-4}$ | 2913 | 54% |
| 50% | $2.9 \times 10^{-4}$ | 2098 | 39% |
| 25% | $1.5 \times 10^{-4}$ | 1047 | 19% |
| 10% | $5.8 \times 10^{-5}$ | 473 | 9% |
| 5% | $2.9 \times 10^{-5}$ | 266 | 5% |

[a] Total number of eQTL-meQTL pairs tested for co-localization was 5397

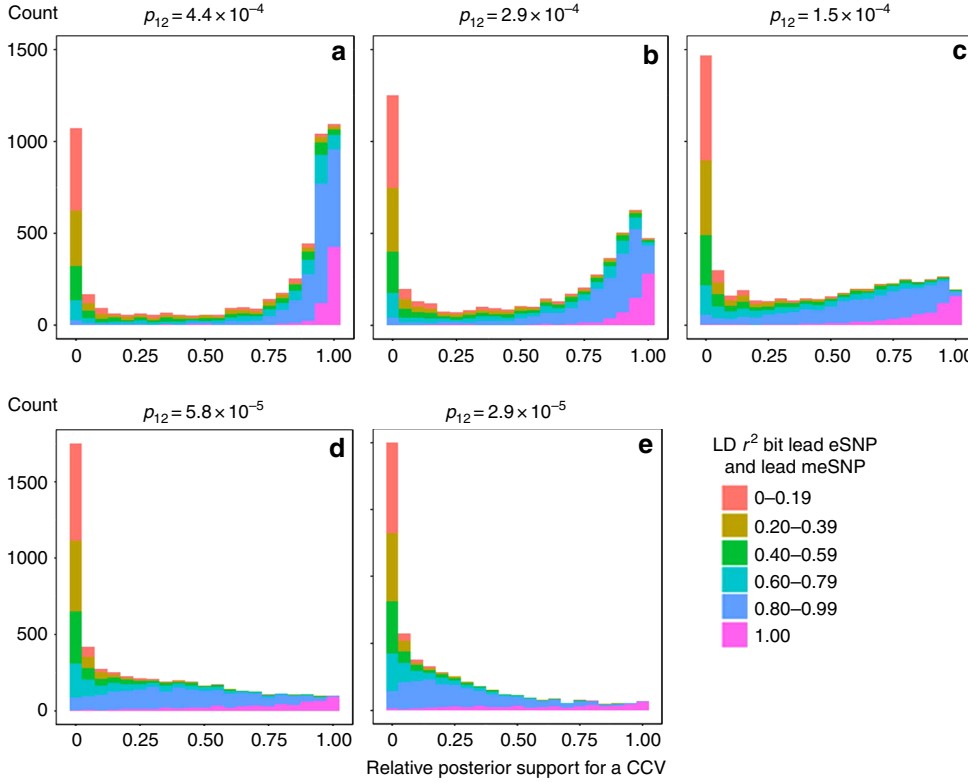

**Fig. 2** Histogram of the relative posterior support for a CCV stratified by strength of LD ($r^2$) between the lead eSNP and lead meSNP. The relative posterior support for a CCV is defined as the posterior probability of a CCV divided by the sum of the posterior probabilities for a CCV and for distinct causal variants (DCV). These data are restricted to co-localization tests for which the sum of the posterior probabilities for DCV and CCV are >0.8 (100 out of 5397 tests excluded). Results are shown for five values of the prior $p_{12}$ (panels **a**-**e**)

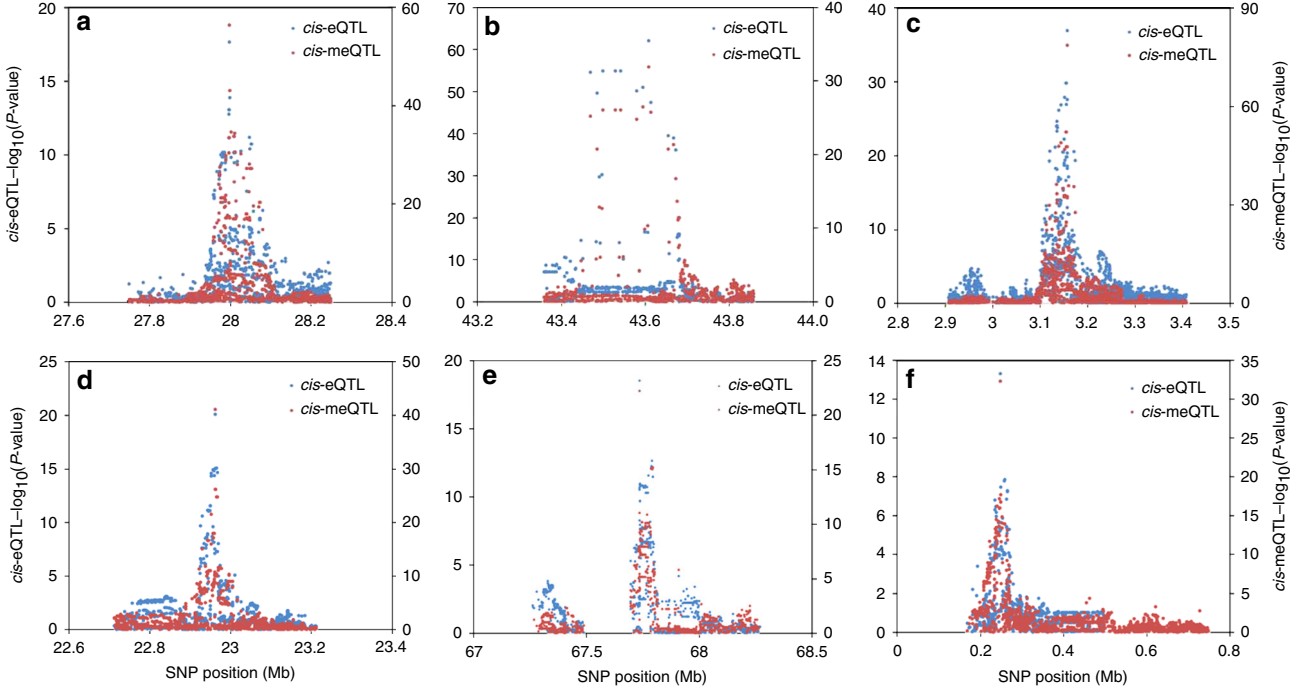

**Fig. 3** Examples of six co-localized eQTL-meQTL pairs. *P*-values for the co-localizing eQTL (blue) and meQTL (red) are plotted against physical position. **a** ILMN_1658464 (*GTF3A*) and cg22138327. **b** ILMN_1694711 (*MAD2L1BP*) and cg14302083. **c** ILMN_1721978 (*CARD11*) and cg19214707. **d** ILMN_1737918 (*C1QA*) and cg10916651. **e** ILMN_2193591 (*UNC93B1*) and cg20272935. **f** ILMN_2282366 (*IQSEC3*) and cg10356759

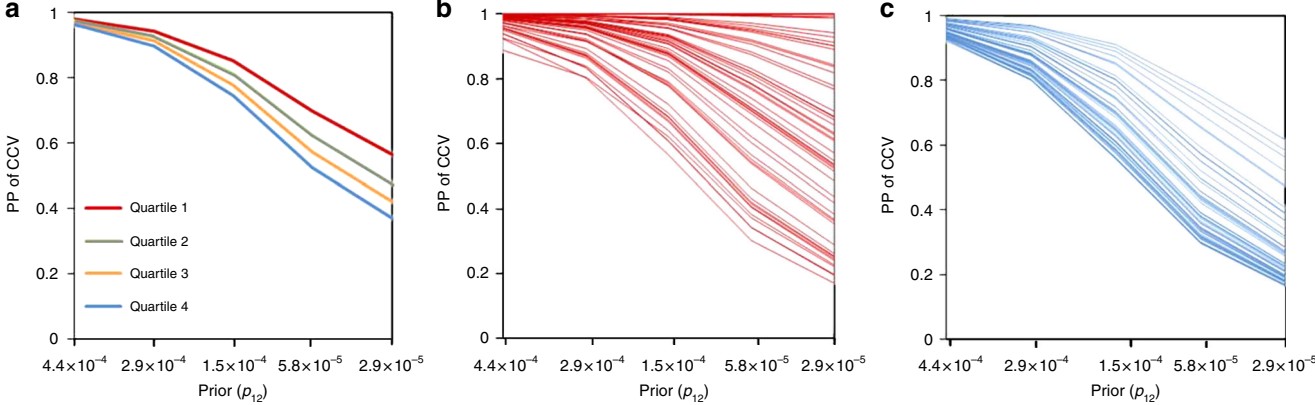

**Fig. 4** The posterior probability (PP) of sharing a common causal variant (CCV) depends on the extent of LD in the region. **a** Average PPs of CCV at different priors ($p_{12}$) stratified by quartiles of LD score of the region. **b** PP of CCV at different prior ($p_{12}$) for the lead eSNPs with the lowest LD score (50 SNPs). **c** PP of CCV at different prior ($p_{12}$) for the 50 lead eSNPs with the highest LD score (50 SNPs)

pairs (522 pairs are involved CpG sites measured on the 450K array but not on the EPIC array). Evidence of co-localization is consistent for the vast majority of the eQTL-meQTL pairs tested for replication (Supplementary Fig. 5).

**Co-localization and LD**. It has been suggested that local LD patterns affect the posterior probability of CCV[20]. Using our co-localization results, we tested the association between the probability of a CCV for each probe-CpG pair and the "LD score"[22] for each lead eSNP. The LD score was defined as the sum of the pairwise $r^2$ values between the lead eSNP and all SNPs within 500 kb (using LD data from unrelated Bangladeshi individuals) and represents the extent to which a SNP is correlated with nearby variants, capturing both strength of LD and quantity of correlated SNPs. We observed a strong inverse association between the LD score and the probability of CCV (Fig. 4a and Supplementary

Data 2), demonstrating the increased uncertainty regarding sharing of a causal variant in regions containing many highly correlated variants. Co-localized eQTL-meQTL pairs with high LD scores for the lead SNPs also appear to have probabilities of CCV values that were more sensitive to the prior ($p_{12}$) than pairs with low LD scores (Fig. 4b, c), indicating that the test for co-localization is better able to detect evidence of a shared causal in regions of low LD. The probability of CCV was also strongly associated with the strength of the eQTL and meQTL association signals (in terms of *P*-value; Supplementary Data 2).

**Partial correlation analysis**. We restricted analyses to 316 genotyped individuals with both expression and methylation data and conducted partial mediation analysis to determine if there was residual correlation between the expression probe and the CpG probe after removing the variance attributable to the lead

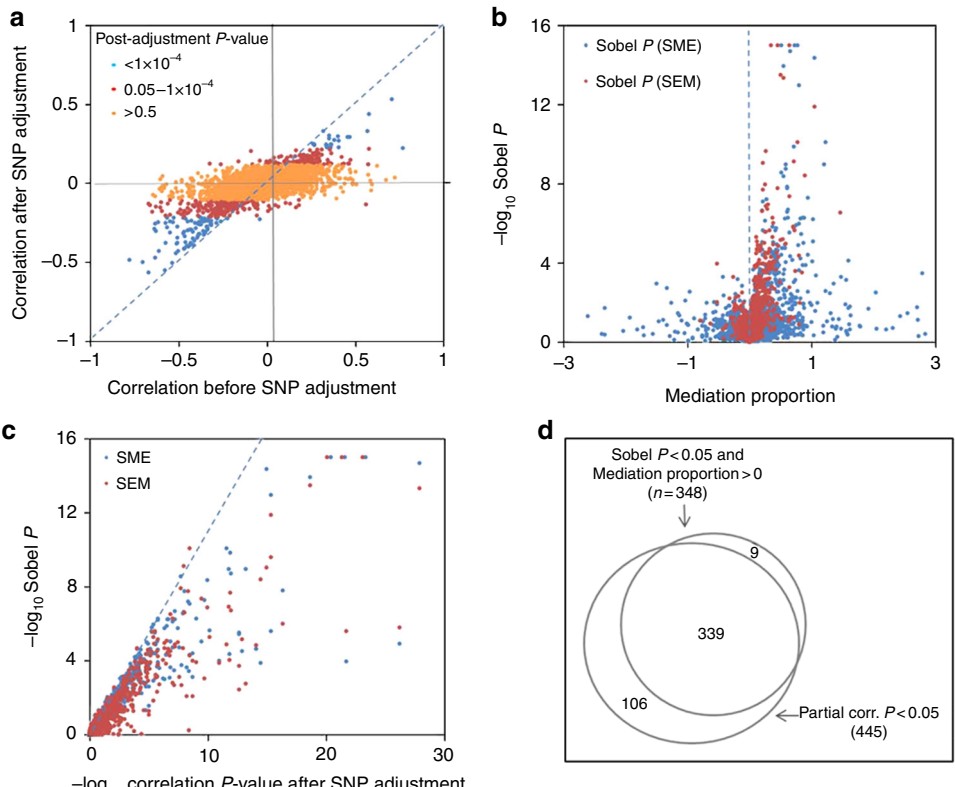

**Fig. 5** Partial correlation and mediation analyses provide evidence for shared regulatory mechanisms. Results for 2913 potentially co-localized eQTL-meQTL pairs identified using a $p_{12}$ value $4.4 \times 10^{-4}$ are presented (results for additional $p_{12}$ values are presented in Supplementary Figures 6–9). All models include adjustments for age, sex, and PCs from both the expression and methylation data ($n = 316$). **a** Results from partial correlation analysis. **b** Mediation analysis results for both the SME (blue) and SEM (red) models. Mediation proportion outliers out of $-3$ to 3 ranges were removed from the figure and Sobel $P$ outliers $<10^{-15}$ were set to the $10^{-15}$. **c** Relationship between Sobel $P$ from mediation analyses and the post-adjustment correlation $P$-values from partial correlation analysis. Sobel $P$ outliers $<10^{-15}$ were set to the $10^{-15}$. **d** Venn diagram showing the concordance between mediation analysis and partial correlation analyses

eQTL SNP (i.e., regressing the probe on the SNP and taking the residual). When a SNP has independent associations with expression and methylation (i.e., pleiotropy, with no causal relationship between expression and methylation), the residuals after removing the variance attributable to the lead SNP would be uncorrelated. As such, observing correlations in residuals provides support for a causal relationship between expression and methylation[19].

Among 2913 potentially co-localized pairs (based on $p_{12} = 4.4 \times 10^{-4}$), we observed 216 eProbe-CpG pairs (7%) showing significant correlation after adjustment for the lead SNP using an FDR of 0.05 and 445 pairs (15%) showing partial correlation with $P < 0.05$. Correlations tended to be weaker in magnitude after SNP adjustment, and significant correlations were more likely to be negative than positive (Fig. 5a). Results for all partial correlation tests are shown in Supplementary Data 3. Results using different values for the prior $p_{12}$ ($2.9 \times 10^{-4}$, $1.45 \times 10^{-4}$, $5.8 \times 10^{-5}$, and $2.9 \times 10^{-5}$) showed similar patterns (Supplementary Figs. 6–9, panel a).

To explore the extent to which partial correlation could be due to secondary, co-localized causal variant affecting both the expression trait and the CpG being analyzed, we searched for secondary association signals for our 445 eQTL-meQTL pairs with partial correlation $P < 0.05$. We identified 83 pairs that had both a secondary eQTL and meQTL, and 31 of these pairs had a probability of CCV > 0.80. Among these pairs, 10 were no longer significant ($P < 0.05$) after adjusting for both the primary and secondary lead eSNP-meSNP (Supplementary Data 4), suggesting

that a small fraction of our findings are due to co-localization of a secondary eQTL-meQTL pair.

**Mediation analysis.** We applied mediation analysis, as previously described[23], to our 2913 potentially co-localized pairs in order to assess evidence that (1) local DNA methylation mediates the effect of a SNP on local gene expression (SNP -> Methylation -> Expression or "SME") and (2) gene expression mediates the effect of a SNP on local DNA methylation (SNP -> Expression -> Methylation or "SEM"), a scenario in which DNA methylation is reactive to variation in gene expression activity. Using an FDR of 0.05, we observed 161 eProbe-CpG pairs showing evidence of mediation under the SME model (Sobel $P < 0.0035$ and % mediation > 0), and 125 pairs showing evidence of mediation under the SEM model (Sobel $P < 0.0024$ and % mediation > 0). The two sets are largely overlapping, with 119 pairs showing mediation under both models, and 167 showing evidence of mediation under at least one model (Fig. 5b). In other words, evidence for mediation was often detected for specific gene-CpG pairs regardless of which causal model was tested (SEM or SME; Supplementary Fig. 10). This demonstrates an important limitation of mediation analysis: while it can be useful for detecting evidence of a causal relationship, mediation analysis can be inadequate by itself for determining the direction of that causal relationship. However, we demonstrate using simulated data that evidence for mediation should be stronger when the causal model (SEM or SME) is correctly specified, even in the presence of

measurement error (Supplementary Fig. 11). In addition, statistical support for a correctly specified model will be more pronounced when the effects along the mediation pathway are stronger (Supplementary Fig. 11). Lack of evidence of mediation implies either pleiotropy (i.e., no causal relationship between expression and methylation) or lack of power to detect mediation. Results for all mediation tests are presented in Supplementary Data 3 and results using different priors ($p_{12} = 2.9 \times 10^{-4}$ and $p_{12} = 1.45 \times 10^{-4}$) show similar patterns and are shown in Supplementary Figs. 6–9 (Panel b).

**Comparison of partial correlation and mediation results.** The mediation analysis results were highly consistent with the partial correlation analysis results. The distribution of the Sobel P-value and the post-adjustment P-value were similar, in that pairs with small Sobel P-values tended to have small P-value for correlation after SNP adjustment (Fig. 5c), with nearly all of the 348 "mediated" pairs ($P < 0.05$, Sobel test) being among the 445 pairs with a significant partial correlation ($P < 0.05$, Pearson correlation) after SNP adjustment (Fig. 5d). Results were similar when

using different values for the co-localization prior $p_{12}$ ($2.9 \times 10^{-4}$, $1.45 \times 10^{-4}$, $5.8 \times 10^{-5}$, and $2.9 \times 10^{-5}$; Supplementary Figs. 6–9, Panels c and d). The proportion of co-localized pairs showing evidence of mediation and/or partial correlation ($P < 0.05$) was 16%, 18%, 24%, 19%, and 17% for the priors $2.9 \times 10^{-4}$, $1.45 \times 10^{-4}$, $5.8 \times 10^{-5}$, and $2.9 \times 10^{-5}$, respectively.

Our ability to detect significant evidence of mediation and partial correlation was strongly related to the strength of the eQTL and meQTL being tested (Supplementary Table 1), indicating that we are likely underpowered to detect mediation and/or partial correlation for a substantial number to eQTL-meQTL pairs. In addition to partial correlation and mediation analyses, we also conducted Bayesian network analysis (BNA) as previously described[17] to determine if either the SME or SEM models were more strongly supported by the data than a model based on independent SNP effects on expression and methylation (see Methods). BNA identified SEM or SME to be the most likely model for 434 pairs with 237 of the pairs also identified by either partial correlation and/or mediation analysis (Supplementary Fig. 12). For, the vast majority of these pairs (389), SME was the most likely model, which was largely consistent with results from

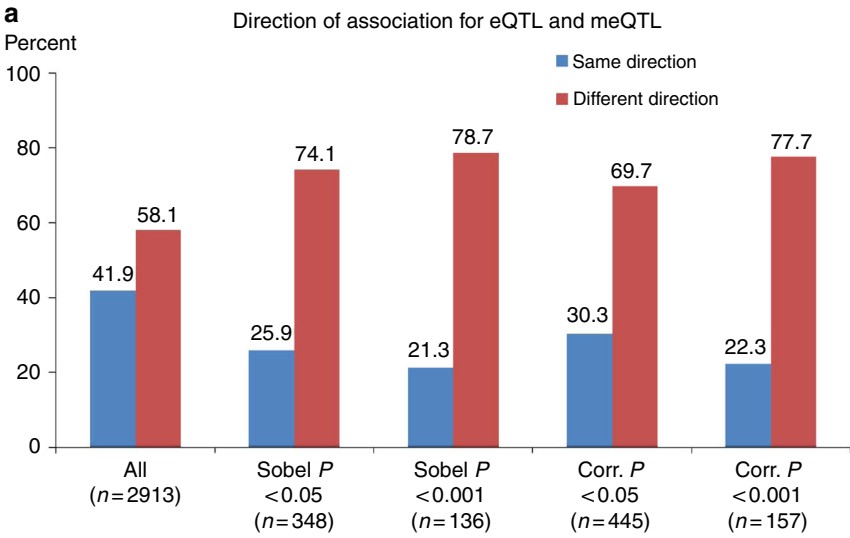

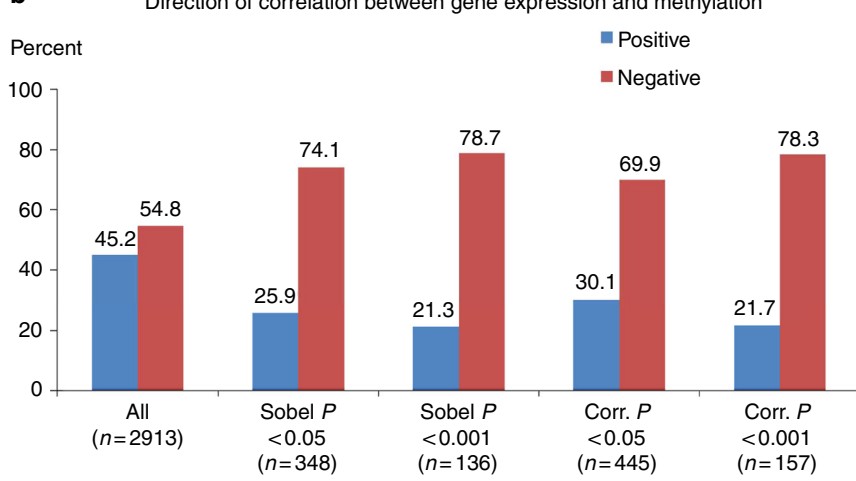

**Fig. 6** Direction of QTL effects and associations between expression and methylation for co-localized eQTL-meQTL pairs. Results for 2913 potentially co-localized eQTL-meQTL pairs identified using a $p_{12}$ value $4.4 \times 10^{-4}$ are presented (results for additional $p_{12}$ values are presented in Supplementary Figures 13-16). Results are stratified according to P-values from mediation analysis (Sobel P) and partial correlation analysis (Corr. P). **a** Histograms of the percentage of eQTL-meQTL pairs showing the same or different direction of association. **b** Histograms of the percentage of eQTL-meQTL pairs for which the direction of association between gene expression and DNA methylation is positive or negative

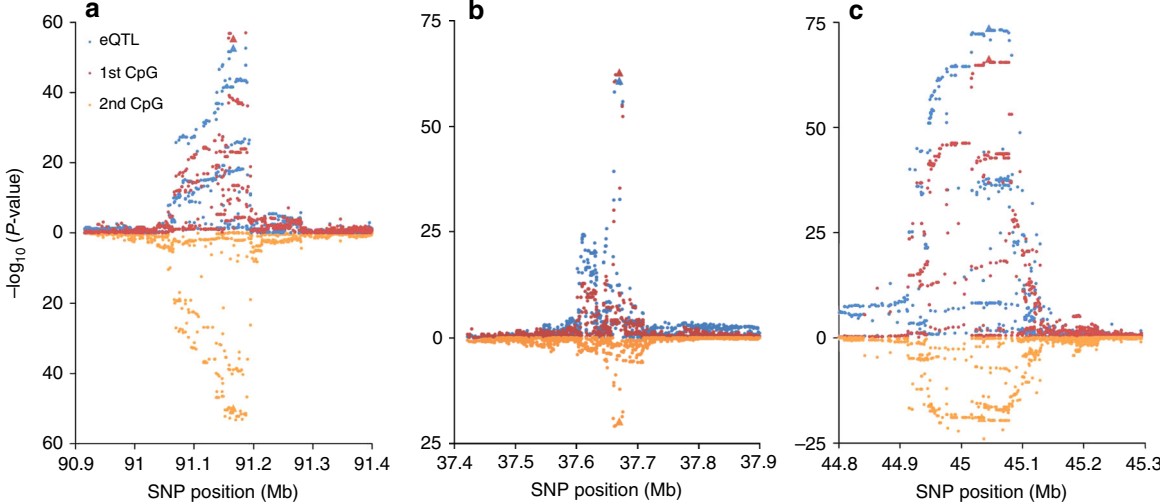

**Fig. 7** Examples of eQTLs that co-localize with a meQTL that has opposite effects on two nearby CpG sites. Results for the CpG selected for co-localization analysis (red) are shown on the top of each plot (ascending), overlaid with the eQTL results (blue). For these scenarios, the expression-increasing allele is associated with increased methylation at a "primary" CpG. Results for a "secondary CpG" are shown below (descending) in orange, for which the expression-increasing allele is associated with decreased methylation. The lead eSNP is shown as a triangle. **a** eSNP: 10:91186842, gene expression probe: ILMN_1696654, first CpG: cg27582166, second CpG: cg13172359. **b** eSNP: 6:37669641, gene expression probe: ILMN_1720595, first CpG: cg26720545, second CpG: cg26129310. **c** eSNP: 3:45047987, gene expression probe: ILMN_2055477, first CpG: cg25593573, second CpG: cg06117855

mediation analysis (Supplementary Fig. 12). Some consistency is expected as both methods rely on conditional dependence. Similar patterns were observed when using different values for the co-localization prior $p_{12}$ ($2.9 \times 10^{-4}$, $1.45 \times 10^{-4}$, $5.8 \times 10^{-5}$, and $2.9 \times 10^{-5}$; Supplementary Fig. 12).

**Co-localized eQTL-meQTL pairs tend to have opposite effects**. Among our 3453 co-localized eProbe-CpG pairs, the direction of the association of the SNP with expression and methylation was more often in opposite directions (58.1%) than in the same direction (41.9%; Fig. 6a), consistent with the hypothesis that reduced promoter methylation is indicative of a more open chromatin state and increased transcriptional activity. When restricting to pairs that show evidence of a shared mechanism, according to either partial correlation or mediation (at either $P < 0.05$ or $P < 0.001$), a more striking difference is observed, with 70–80% of co-localized eQTL-meQTL pairs showing opposite directions of association, depending on the $P$-value threshold used (Fig. 6a). Similarly, the expression and methylation traits for co-localized pairs were more often negatively correlated than positively correlated (55% negative). This imbalance was much stronger after restricting to pairs showing evidence of a shared causal mechanism, according to either partial correlation or mediation analysis, with 70–80% of co-localized eQTL-meQTL pairs showing an inverse correlation (based on $P < 0.05$ and $P < 0.0001$; Fig. 6b). Similar patterns were observed when using different values for the co-localization prior $p_{12}$ ($2.9 \times 10^{-4}$, $1.45 \times 10^{-4}$, $5.8 \times 10^{-5}$, and $2.9 \times 10^{-5}$; Supplementary Figs. 13–16).

**Expression-increasing alleles can both increase and decrease local methylation**. There were 1457 co-localized eQTL-meQTLs pairs showing associations with expression and methylation in the same direction (i.e., an allele is associated with an increase in both expression and methylation). Among these meQTLs, we searched for additional nearby CpG sites that showed an association with the SNP that was in the opposite direction of the eQTL. In 955 out of 1457 cases, we identified at least one such a secondary CpG, and these CpGs were consistently inversely associated with the CpG originally selected. In other words, many

of our eSNPs/meSNPs were associated with methylation at multiple nearby CpGs, with the minor allele increasing methylation at one CpG while decreasing methylation at another. The three examples with the strongest association between SNP and secondary CpG are shown in Fig. 7, and all of these secondary meQTL signals also co-localize with the primary eQTL with probabilities of CCV > 90%. The distribution of the distance between the primary and secondary CpGs observed was <100 kb in ~75% of cases and is shown in Supplementary Fig. 17.

**Examples of co-localized eQTLs-meQTLs with strong mediation**. For several eQTL-meQTL pairs with strong evidence for co-localization and mediation, we examined potential biological mechanisms based on genomic annotations. SNP rs2069235 (chr22:39747780) was the lead *cis*-eSNP for expression probe ILMN_1810875 ($P = 10^{-115}$), which captures almost all isoforms of *SYNGR1*[21]. This eQTL co-localized with a meQTL associated with methylation at seven CpG sites: cg24268161; cg21658277; cg07919145; cg20496314; cg22247277; cg22628235; and cg22650271 (Fig. 8 and Supplementary Fig. 18). For the eQTL-meQTL pair, we observed strong evidence of partial correlation (residual $r = -0.29$; $P = 9 \times 10^{-8}$) and mediation (SEM $P = 2 \times 10^{-8}$; SME $P = 3 \times 10^{-7}$). SNP rs2069235 is in strong LD ($r^2 > 0.65$) with only two SNPs rs909685 ($r^2 = 1.0$) and rs9611155 ($r^2 = 0.89$) based on 1KG BEB data. SNPs rs2069235 and rs909685 reside within 109 bp of each other, and both overlap with multiple TF-binding sites, broad peaks for both H3K4Me1 and H3K27Ac, and rs2069435 overlaps with a 51-cell/-tissue DNase I hypersensitivity site (Fig. 8), suggesting one of these SNPs may disrupt TF binding, which would imply the SEM model as a biological mechanism. SNP rs909685 also resides only ~200 bp from cg24268161, the CpG showing the strongest meQTL association ($P = 10^{-107}$), suggesting meQTLs effects mediated by expression may be stronger for CpGs very close to the relevant binding site. SNPs rs2069235 and rs909685 have been reported as risk factors for primary biliary cirrhosis[24] and rheumatoid arthritis[25], respectively, suggesting the CCV has pleiotropic effects on these phenotypes. SNP rs9611155 showed little overlap with the annotations examined. Two additional

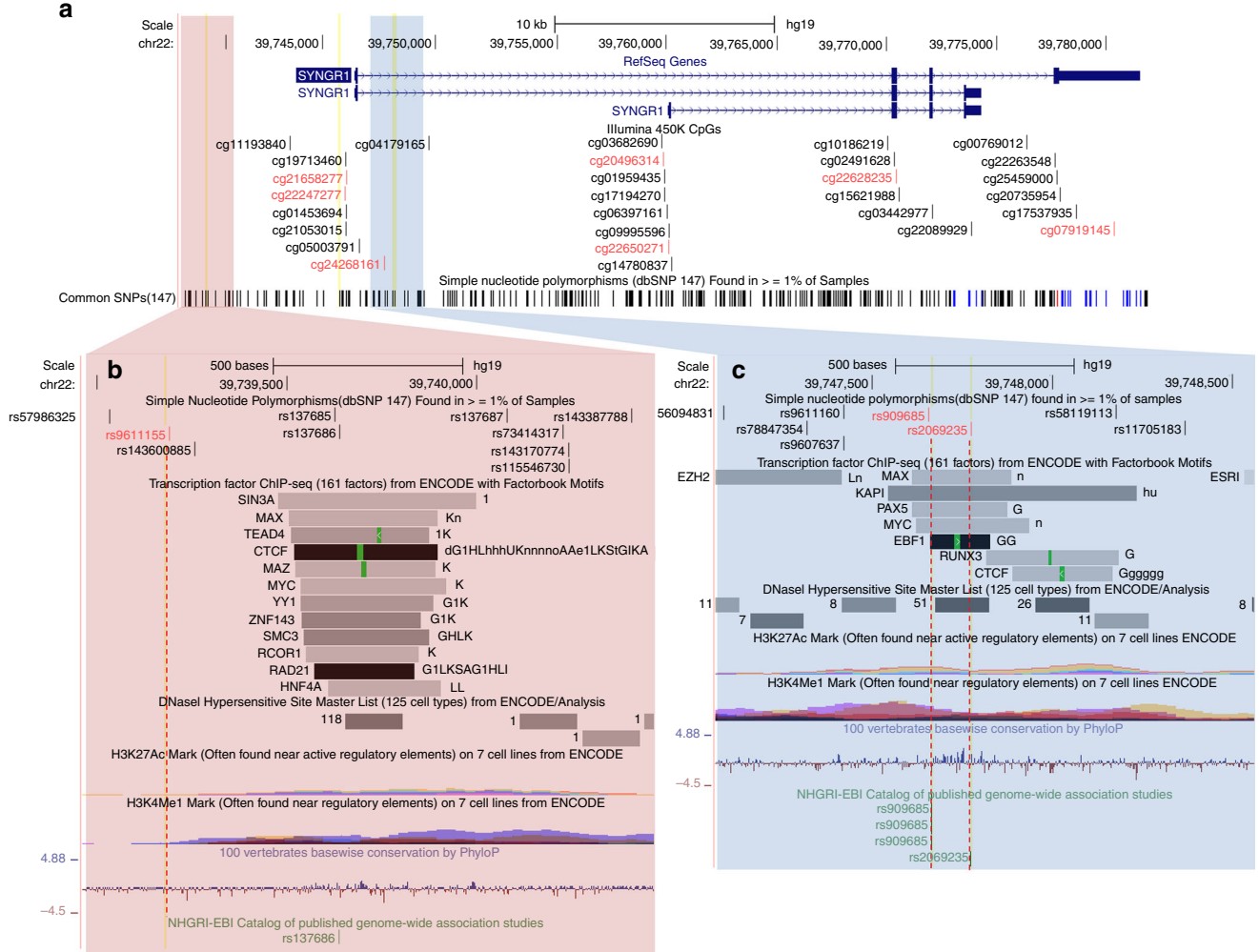

**Fig. 8** Overlap with genomic annotations for three candidate causal SNPs for an eQTL that co-localizes with seven meQTLs and shows strong evidence of mediation. **a** The *SYNGR1* gene region has seven CpG sites (in red) affected by a common causal variant. Possible causal variants include rs9611155 (**b**) and rs909685 and 2069435 (**c**)

examples of co-localized eQTL-meQTL pairs with strong evidence of mediation are described in Supplementary Figs. 19 and 20, but neither of which strongly support a single SNP as a disruptor of TF binding.

## Discussion

In this work, we have described the extent to which peripheral blood eQTLs and meQTLs share common causal variants using data from a Bangladeshi cohort. We identified a set of 2913 eQTL-meQTL pairs that potentially share a CCV, although this number is sensitive to the priors used in co-localization analysis. We used partial correlation analysis and mediation analysis to assess the evidence that these pairs of expression and methylation traits are causally related to one another, sharing a common biological mechanism. Among the 2913 potentially co-localized pairs, we found such evidence for >400 pairs, with mediation and partial correlation analysis showing highly consistent results. The proportion of co-localized pairs showing evidence of mediation (and/or partial correlation) was fairly consistent regardless of the prior used, varying between 15 and 24%. We expect there are many additional examples of mediation that were are under-powered to detect. Our results demonstrate common co-localization of eQTLs and meQTLs in the human genome and shared biological mechanisms. The approach taken here can be

extended to other types of cellular/molecular QTLs (e.g., SNPs affecting chromatin features, protein abundance, etc.) in order to enhance our understanding of the cascade of regulatory mechanisms by which SNPs can affect gene expression and function, which is critical for understanding how SNPs affect human disease.

The SNP underlying each co-localized eQTL-meQTL pair tended to have opposite effects on expression and methylation, consistent with the view that hypo-methylation near the promoter and the transcription start site reflects accessible chromatin, binding of TFs, and active transcription. Prior studies have suggested that one of the mechanism underlying co-localized pairs is disruption of TF-binding sites, which can reduce TF-binding affinity, thereby reducing transcriptional activity in the gene region and producing "reactive" changes in chromatin structure (including local DNA methylation)[15, 19]. Little is known regarding mechanisms by which genetic variation directly affects DNA methylation (and other chromatin features indicative of chromatin conformation), which in turn impacts expression. Interestingly, there were no DNA methyltransferases (e.g., *DNMT1, DNMT3,* and *TRDMT1*) among the gene pairs classified as SEM (or SME). However, recent work suggests new scenarios in which DNA methylation can create new binding sites for transcription factors, potentially leading to alternative binding sites in the presence of high levels of methylation[26]. This is an

interesting possibility, considering a subset of our co-localized eQTL-meQTL pairs appear to (a) affect expression and methylation in the same direction and/or (b) affect multiple CpG sites in opposite directions, suggesting a more nuanced relationship between DNA methylation and local gene expression.

Several prior studies have attempted to assess causal relationships among expression and methylation features that are associated with a common SNP[15–18]; however, in this study we use co-localization methods to identify relevant eQTL-meQTL pairs. Co-localization is critical for selecting eQTL-meQTL pairs for analyses focused on understanding causal relationships, as such analysis make the implicit assumption that eQTLs and meQTLs share common causal variant. The LD between the lead SNPs for the two QTLs is predictive of the probability of a CCV, but co-localization analysis allows for quantification of the uncertainty regarding the probability that an eQTL and meQTL share a CCV. For a specific meQTL, choosing a CpG remains a challenge, as a meQTL can be associated with increased methylation at one CpG and decreased methylation at another, and both can co-localize with an eQTL. Specifying appropriate priors for co-localization is also a challenge, considering posterior probabilities are sensitive to the choice of priors[21, 27]. However, there are several recently proposed methods that use genome-wide summary statistics for both traits and analyses of enrichment to estimate these priors[28–30], thereby avoiding subjective decisions regarding prior specification. While the "internal empirical calibration" approach we used suggested that $4.4 \times 10^{-4}$ was the best choice for $p_{12}$, the number of instances of co-localization detected was consistently smaller than the prior expectation for "true" co-localized pairs for all five values of $p_{12}$, with smaller discrepancies observed for smaller values for $p_{12}$ (Table 2). These discrepancies may be due in part to limited statistical power for co-localization analysis of weak QTL signals or may reflect a limitation of the internal empirical calibration approach.

We focus on two approaches for examining evidence that co-localized eQTL-meQTL pairs represent causal SNPs that effect both expression and methylation along a common causal pathway. The first method, partial correlation analysis, detects correlation between expression and methylation that is independent of the regulatory SNP (i.e., residual correlation after adjusting both phenotypes for the SNP). Lack of correlation after adjustment suggests there is not a causal relationship between methylation and expression, as correlation is purely driven by SNP effects[19]. The second method, mediation analysis, is a test for shared phenotypic variance amongst the SNP, transcript, and methylation. Mediation analysis can also be conceived of as a test for attenuation of the SNP-phenotype relationship after adjusting for a potential mediator. Mediation can be viewed as a more stringent test than partial correlation, as the presence of mediation implies non-zero partial correlation. For both of these methods, we must keep in mind that there are limitations for all tests used to assess evidence of causality; and these tests cannot be used as definitive evidence of causality for any given eQTL-meQTL pair. For example, for some pairs there could exist hidden confounders that are not well captured by the principal components (PCs) variables we adjust for, and the presence of mediator-outcome confounding can introduce bias into mediation analyses[31]. In addition, while our Sobel P-values tend to favor the SME model over the SEM, we cannot determine the direction of causality for any given pair of expression and methylation traits that appear to be causally related to one another. In other words, for any given pair, it is possible that (a) hyper-methylation near the transcription start site makes DNA less accessible for TF binding or (b) binding site polymorphisms affect transcription initiation, which then in turn affects chromatin structure, including DNA methylation.

A substantial proportion of our eQTLs do not co-localize with a meQTLs, and several factors may contribute to this observation. First, the co-localization analysis approach we use estimates the probability of a single common causal variant and is not a test for multiple causal variants. Thus, it is possible that the presence of multiple causal variants or non-shared causal variants near a shared causal variant may reduce power to detect co-localization of a shared variant or variants. Recently developed methods can address this issue[27]. Second, we are likely underpowered to detect co-localization when the eQTL and/or eQTL associations are weak, as the probability of a CCV clearly depends on the strength of the association. Third, the probability of a CCV was systematically lower in high LD regions, making it less likely to detect true co-localization in such regions. Fourth, the RNA and DNA samples used for expression and methylation measurements were not obtained from identical populations of white blood cells (mononuclear cells vs. whole blood, respectively). peripheral blood mononuclear cells (PBMCs; monocytes, T lymphocytes, and B lymphocytes) account for ~35% of peripheral white blood cells; thus, the remaining 65% of peripheral white blood cells (neutrophils, basophils, and eosinophils) are represented in our DNA methylation data but not in our expression data. Thus, for co-localized eQTL-meQTL pairs that are specific to PMBC subtypes, the meQTL signal may be weak in our data due to the presence of the many cell types in whole blood that are not PBMCs. Lastly, it is likely that only a subset eQTLs impact DNA methylation. In lymphoblastoid cell lines, for example, it has been reported that only 10–20% of eQTLs are also meQTLs[4]. eQTL mechanisms that would not necessarily involve local epigenetic alterations include effects on mRNA processing or mRNA stability[19].

Depending on what priors were used for co-localization analysis, only 15–24% of co-localized pairs showed evidence of mediation and/or partial correlation, and this apparent discrepancy may be due, at least in part, to several factors that reduce statistical power. First, our mediation and partial correlation analyses are likely underpowered for many of these tests, which require participants with both expression and DNA methylation data. We have only 316 such individuals. In light of the strong association we observe between the strength of the eQTL and meQTL associations and the P-values for our tests of mediation and partial correlation, power is likely to be low for many tests. Second, the cell type issue above will also reduce power to detect mediation and partial correlation, as we are analyzing a mixture of cell types in the presence of cell-type-specific QTLs. The proportion of eQTLs that strongly co-vary with meQTLs in this data set may be lower than would be observed in a study of similar size focused on a specific cell type, as our methylation measures capture variation in methylation attributable to many different cell types. Third, considering all transcripts and CpGs are imperfect measures, and the CpG we select for analysis is a proxy for some underlying epigenetic state, our power is likely reduced by measurement error. In fact, much of the mediation evidence we detect is "partial mediation" (i.e., mediation proportion < 1), and we have shown that this is expected when full mediation is present, but the mediation measure is error-prone[23].

This work demonstrates the pervasiveness of co-regulated expression and methylation traits in the human genome. Future studies should develop methods for combining data on clustered CpG sites to characterize the effects of SNPs on local methylation and their implication for local chromatin structure. This is important as most meQTLs are associated with methylation at multiple CpGs, sometimes in opposing directions. Future studies should also apply this approach to other types of molecular QTLs, including additional indicators of chromatin structure, such as

histone features, to enhance our understanding of regulatory mechanisms.

## Methods

**Study population**. Subjects included in this work were participants in the BEST[32]. BEST is a randomized chemoprevention trial evaluating the long-term effects of vitamin E and selenium supplementation on non-melanoma skin cancer risk among 7000 individuals with arsenic-related skin lesions living in seven sub-districts in Bangladesh. BEST participants were 54% male, with a mean age of 43 years. Participants included in this work are a subset of BEST participants from Araihazar that have available data on genome-wide SNPs and array-based expression and DNA methylation measures (described below). For replication of co-localization results, we used data on 347 participants from the HEALS[33]. The HEALS and BEST studies and associated genomics research were approved by the Institutional Review Boards of The University of Chicago, Columbia University, and the Bangladesh Medical Research Council, and all study participants provided informed consent.

**Genotyping and imputation quality control**. DNA extraction for genotyping was carried out from the whole blood using the QIAamp 96 DNA Blood Kit (catalog # 51161) from Qiagen, Valencia, USA. Concentration and quality of all extracted DNA were assessed using Nanodrop 1000. As starting material, 250 ng of DNA was used on the Illumina Infinium HD SNP array according to Illumina's protocol. Samples were processed on HumanCytoSNP-12 v2.1 chips with 299 140 markers and read on the BeadArray Reader. Image data were processed in BeadStudio software to generate genotype calls.

Quality control (QC) was conducted as described previously for a larger sample of 5499 individuals typed for 299 140 SNPs[34, 35]. We removed DNA samples with call rates <97% ($n = 13$), gender mismatches ($n = 79$), as well as technical duplicates ($n = 53$). We removed SNPs that were poorly called (<90%) or monomorphic ($n = 38\,753$), and then removed SNPs with call rates < 95% ($n = 1045$) or Hardy–Weinberg equilibrium (HWE) $P$-values < $10^{-10}$ ($n = 634$, which produces no HWE (exact) $P$-values < $10^{-7}$ in a subset of 1842 unrelated participants). This QC resulted in 5354 individuals with high-quality genotype data for 257 747 SNPs. The MaCH software[36] was used to conduct genotype imputation using 1000 genomes reference haplotypes (1KG phase3 v5, which includes South Asian populations). Only high-quality imputed SNPs (imputation $r^2 > 0.5$) with SNPs with minor allele frequency > 0.05 were included in this analysis. A subset of 1329 unrelated individuals with available data on array-based expression and DNA methylation measures was used for this project. Only autosomal SNPs were included in this analysis.

**DNA methylation**. BEST DNA samples were extracted from whole blood using DNeasy Blood kits (Qiagen). Bisulfite conversion was performed using the EZ DNA Methylation Kit (Zymo Research, Irvine, CA, USA). For each sample, 500 ng of bisulfite-converted DNA was applied to the Illumina HumanMethylation 450K BeadChip kit (Illumina, San Diego, CA, USA) according to the manufacturer's protocol, enabling interrogation of 482 421 CpG sites and 3091 non-CpG sites per sample. This array contains an average of 17 CpG sites per gene, distributed across the promoter, 5′ untranslated region (UTR), first exon, gene body and 3′ UTR, covering 99% of RefSeq genes. DNA methylation data for replication purposes were obtained from HEALS. HEALS DNA samples were extracted from clot blood using Qiagen Flexigene DNA kits (catalog # 51204), and DNA methylation was measured using the Illumina MethylationEPIC array, which measures >90% of the CpG sites measured by the 450K array.

Methylation status at each CpG is expressed as a $\beta$ value that can range from 0 (unmethylated) to 1 (completely methylated). Data were quantile normalized. Among the 413 participants, we excluded 6 samples for which the reported sex of the participant did not correspond with predicted sex based on methylation patterns of the X and Y chromosomes, and 7 samples with >5% of CpGs either containing missing values or having $p$ for detection > 0.05. This resulted in 400 samples with quality methylation data. We removed probes mapping to multiple locations (41 937) and probes with SNPs (20 869) according to Price et al.[37] Individual $\beta$ values with a $p$ for detection > 0.05 were set to missing, and we excluded probes if >10% of beta values were missing (1636). We also excluded probes on the X (11 232) and Y (416) chromosomes, probes with missing chromosome data (mostly control probes, 65), and probes with >10% missing data across samples (1932); this resulted in a total of 423 604 probes available for analysis. $\beta$ values were logit-transformed and adjusted for batch variability using ComBat software[38]. Based on 11 samples run in duplicate across two different plates in these experiments, the average inter-assay Spearman correlation coefficient was 0.987 (range, 0.974–0.993). For the HEALS EPIC array data, we removed 1 individual due to sex mismatch, 10 due to lack of GWAS data, and 42 due to cryptic relatedness. We removed 26 629 probes with detection $P > 0.01$ in one or more samples, 85 probes missing >5% data, 41 920 cross-reactive probes, 7791 probes with SNP at target CpG site or within single-base extension, 57 rs probes, 2460 non-CpG probes, and 16 761 probes on X and Y chromosome. After QC, we have data for 347 individuals and 771 192 CpGs.

**Gene expression**. RNA was extracted from PBMCs, preserved in buffer RLT, and stored at −86 °C using RNeasy Micro Kit (catalog # 74004) from Qiagen. Concentration and quality of RNA samples were assessed on Nanodrop 1000. cRNA synthesis was done from 250 ng of RNA using Illumina TotalPrep 96 RNA Amplification kit. As recommended by Illumina we used 750 ng of cRNA on HumanHT-12-v4 for gene expression. Expression data were quantile normalized and $\log_2$ transformed. The chip contains a total of 47 231 probes covering 31 335 genes. There were 1825 unique individuals in both expression data and SNP data. For the vast majority of participants, between 30 and 47% of probes had detection $P$-values < 0.05. However, 26 individuals had <30% of probes with detection $P$-value < 0.05, and these outlying individuals were excluded from the analysis, leaving an analysis sample size of 1799. For this analysis, no probes were excluded based on detection $P$-values.

**Eligibility for analyses**. The participants and workflow are described in Fig. 1 and Supplementary Fig. 1. Participants included in eQTL analyses included 992 participants with available SNP data and expression data who were unrelated to other participants based on an estimated coefficient of relationship < 0.08. Participants included in meQTL analyses included 337 participants with available SNP data and DNA methylation data who were unrelated to other participants based on an estimated kinship coefficient of <0.08. These samples used for eQTL and meQTL analyses were entirely independent (i.e., non-overlapping participants), which is a requirement for using co-localization methods[20]. Among the 337 participants included in meQTL analyses, 316 of these participants also had expression data (which was not used for eQTL analyses), and these 316 participants were used for mediation analyses, Bayesian network analyses, and partial correlation analyses.

**eQTL and meQTL analyses**. Prior to analysis, expression values were log-transformed and methylation beta values were logit-transformed and adjusted for potential batch/chip effects. Linear regression implemented in the FastQTL software package[39] was used to conduct genome-wide *cis*-eQTL and *cis*-meQTL analyses. *Cis* associations were tested for SNPs and probes <500 kb apart using genotype dosages. For both the *cis*-eQTL and meQTL analyses, adaptive permutations were used in FastQTL (--permute 1000 10000) to obtain beta distribution-adjusted empirical $P$-values. A FDR threshold of 0.01 was applied at the probe level (for both gene expression probe and CpG probes) using the qvalue package in R to identify probes with a significant QTL. SNP-probe pairs with a probe-level $q$-value < 0.01 were defined as a significant eQTL-meQTL. In addition to adjusting for age and sex, we included 80 expression PCs in our eQTL analyses and 10 methylation PCs in our meQTL analyses, and these were selected to maximize the number of *cis* signals detected[23]. No genotyping PCs were included because this is a very homogeneous cohort with no evidence of population strata (as previously reported[34, 35]), with eigenvalues from the first 10 PCs being very similar (between 1.48 and 1.136). Lead eSNPs and meSNPs for each eGene and mCpG, respectively, were defined as the SNP with the smallest $P$-value.

**Identifying QTL pairs likely to share a causal variant**. Our workflow for identifying co-localized eQTL-meQTL pairs (sharing a common causal variant) is shown in Supplementary Fig. 1. For our eQTL results, we first restricted to lead SNPs for each eProbe. Using the meQTL results, we then identified CpGs that were also associated with a lead eSNP. Because clusters of CpGs are often correlated and influenced by the same *cis* variation[40], we pruned our list of CpG probes to reduce this redundancy. We pruned by first identifying CpGs that were associated with the same SNP, and kept only the CpG whose lead meSNP had the highest LD with a lead eSNP. We required each expression probe to be in a pair with only one CpG, the CpG whose lead meSNP was in the strongest LD with the expression probe's lead eSNP. This workflow resulted in eProbe-CpG pairs showing association with a common SNP and able to be tested for co-localization.

**Co-localization analysis**. To assess the probability that *cis*-meQTLs and *cis*-eQTLs residing in the same genomic location were due to the same (single) causal variant, we applied a Bayesian test for co-localization[20] (as implemented in the coloc R package) to all co-occurring eQTL-meQTL pairs, in order to estimate the probability that each QTL pair was due to the same causal variant. This method takes two sets of summary statistics as input. For each eQTL, we used the association results for all SNPs within 250 kb of the lead eSNP. For each corresponding meQTL/CpG, we used results for the same set of SNPs selected from the eQTL results (except SNPs > 500 kb away from the target CpG, as these were not included in *cis*-meQTL analyses). The coloc package only uses information on SNPs present in both sets of summary statistics. The Bayesian co-localization requires specifying a prior probability for a SNP being associated with trait 1 only ($p_1$), trait 2 only ($p_2$), and both traits ($p_{12}$). For the eQTL analysis, we detected 5022 independent eSNPs among 8 639 940 total SNPs, indicating the probability a SNP is a causal eSNP is $5.8 \times 10^{-4}$. This probability corresponds to the sum $p_1 + p_{12}$. For the meQTL analysis, we detected 29 472 independent meSNPs among 8 639 940 total SNPs, indicating the probability a SNP is a causal meSNP is $3.4 \times 10^{-3}$. This probability corresponds to the sum $p_2 + p_{12}$. Thus, our choice for $p_{12}$ impacts the value of $p_1$ and $p_2$. We varied the value of $p_{12}$ ($4.4 \times 10^{-4}$, $2.9 \times 10^{-4}$, $1.45 \times 10^{-4}$, $5.8 \times 10^{-5}$, and $2.9 \times 10^{-5}$) to correspond to probabilities of a causal eSNP being a causal

meSNP of 75%, 50%, 25%, 10%, and 5%, which we view as a large and reasonable range for this prior.

We evaluated the three $p_{12}$ values we used for co-localization using a method described by Guo et al.[21] as "internal empirical calibration". Guo et al. propose that the most appropriate value for $p_{12}$ is the value for which the posterior expectation of co-localization is similar to the prior expectation of co-localization. The results of this analysis are presented as Supplementary Fig. 3.

**Partial correlation analysis**. Using our set of co-localized eQTL-meQTL pairs, we used data on 316 genotyped individuals with both expression and methylation data to conduct partial correlation analysis[19]. We first calculated the Pearson correlation coefficient between the expression probe and the methylation probe (both adjusted for expression and methylation PCs, respectively, as described above). We then regressed both the methylation probe and the expression probe on the lead eSNP, and took the residuals from these regressions to obtain expression and methylation values that lack the phenotypic variance due to the effect of the SNP. We then compare the correlation coefficient before SNP adjustment vs. after SNP adjustment.

**Mediation analysis methods**. Using our set of co-localized eQTL-meQTL pairs, we used data on 316 genotyped individuals with both expression and methylation data to conduct tests of mediation for two hypothesized pathways: (1) "SME" and (2) "SEM". Mediation analysis was conducted as follows: for all lead eSNP, the cis-eQTL association was re-estimated, adjusting for methylation of the CpG (and vice versa). The difference between the beta coefficients before and after adjustment for the cis probe was expressed as the "proportion of the total effect that is mediated" (i.e., % mediation), calculated as $(\beta_{unadj} - \beta_{adj})/\beta_{unadj}$[41], with $\beta_{unadj}$ and $\beta_{adj}$ known as the total effect and the direct effect, respectively. All regressions were adjusted for expression and methylation PCs. The Sobel $P$-value for mediation[42] was calculated by first estimating the cis-eQTL association adjusting for methylation (and vice versa):

$$Y_{cis} = \beta_0 + \beta_{adj}G_{SNP} + \beta_1 X_{cis} + \varepsilon_i \tag{1}$$

We then estimated the eSNP's association with the potentially mediator:

$$X_{cis} = \beta_0 + \beta_2 G_{SNP} + \varepsilon_i \tag{2}$$

The $P$-value was then estimated by comparing this following $t$ statistic to a normal distribution:

$$t = \beta_1\beta_2/\text{SE} \tag{3}$$

$$\text{SE} = \sqrt{\beta_1^2\sigma_{\beta_2}^2 + \beta_2^2\sigma_{\beta_1}^2} \tag{4}$$

where SE is the pooled standard error term calculated from the above beta coefficients and their variances. $\beta_1\beta_2$ is often referred to as the indirect effect.

**Mediation analysis of simulated data**. Using data on a bi-allelic SNP ($G$) for 316 participants (same sample sizes as our analyses), we simulated data on a molecular phenotype ($X$) as a randomly generated standard normal variable with a linear effect exerted by the SNP.

$$x_i = \beta_{GX}g_i + \varepsilon_{Xi} \text{ with } \varepsilon_{Xi} \sim N(0,1) \tag{5}$$

$X$ served as a mediator for the effect of the SNP on a second molecular phenotype ($Y$), which was generated as a standard normal variable with a linear effect exerted by the ($X$).

$$y_i = \beta_{XY}x_i + \rho_{Yi} \text{ with } \rho_{Yi} \sim N(0,1) \tag{6}$$

The variance in the mediator ($X$) explained by the SNP was varied from 0.01 to 0.75. The magnitude of the effect of the mediator on the second molecular phenotype ($\beta_{XY}$) was varied from 0.01 to 0.75. We then used mediation analysis methods described in the section above to obtain a Sobel $P$-value and an estimate of the % mediation. These analyses were conducted in two ways: using $X$ as the mediator and using $Y$ as the mediator.

**Data availability**. Genome-wide summary statistics from the eQTL and meQTL analyses and summary statistics used for co-localization analyses are available at http://datadryad.org/ with the identifier doi:10.5061/dryad.hq68q. Individual-level data used for mediation analysis and partial correlation analysis are accessible via the corresponding author upon reasonable request.

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

## Acknowledgements

We would like to thank all BEST study participants and research staff. This work was supported by National Institutes of Health grants R21ES024834 (B.L.P. and M.A.), R01ES020506 (B.L.P.), R01ES023834 (B.L.P.), R35ES028379 (B.L.P.), R01 GM108711 (L.S.C.), and R01CA107431 (H.A.)

## Author contributions

B.L.P., L.S.C., and H.A developed the general research question. B.L.P. developed the analysis approach, supervised the analyses, and drafted the manuscript. L.T. conducted all statistical analyses. H.A. established BEST and led the collection of all bio-specimens and generation of SNP, expression, and methylation data. M.G.K. and F.J. supervised and conducted DNA and RNA extraction and the generation of SNP, expression, and methylation data. L.S.C. provided assistance with statistical aspects of the work and manuscript preparation. K.D. provided assistance with analyses of the HEALS DNA methylation data. M.A., M.R.-Z., G.S., M.T.I, H.S., T.I., M.R., and M.Y. contributed to BEST study design and data collection. All authors reviewed and edited the manuscript.

## Additional information

**Competing interests:** The authors declare no competing financial interests.

