## [Peer Review File · Nature Communications]

Reviewers' comments:

Reviewer #1 (Remarks to the Author):

Pierce et al. discuss a joint expression QTL and methylation QTL analysis to identify potential common causal variants (CCVs) that impact both expression and methylation phenotypes. The analyses described in the manuscript are mostly carefully conducted, and the details are well documented. I do have a few concerns with respect to the conclusions drawn from the analyses, and I hope my comments are helpful for the authors to revise the manuscript.

- The colocalization analysis needs to be improved. The authors are generally careful in the colocalization analysis using the coloc method. They clearly separate the non-overlapping samples for eQTL and mQTL analyses and perform the sensitivity analysis with respect to the different prior values. Nevertheless, in my opinion, there is a lack of justification why their final results rely on one set of prior (in particular $p_{12} = 1e-5$) vs. the others. Without controlling false positives, it is questionable to reject the more stringent threshold (10^{-7}) based on the fact that it would "eliminate vast majority of evidence for co-localization" (line 153, page 8).

In general, I find that the data presented in this paper do not match the guideline given by the coloc method, largely because the coloc guideline is intended for GWAS where the signals are typically much more sparse. This discrepancy impacts not only the choice of prior parameter p_{12} but also key prior parameters p_1 and p_2 . Specifically based on the total number of interrogated SNPs, unique significant eQTL and mQTL SNPs in Table 1, I estimate p_1 (abundance of eQTLs) should be $\sim 10^{-3}$ (9629/8639940) and p_2 (abundance of mQTLs) should be $\sim 10^{-2}$. These are very different than the values used in the analysis (both are set at 10^{-4})

- The conclusion of the mediation (as well as partial correlation) analysis is very confusingly written (line 208 to 213). The purpose of the mediation analysis seems to be establishing the potential causal model between genetic variant, methylation level, and expression level. The authors should also make clear that pleiotropy is another potential causal model in consideration. After reading the text, my conclusion is that the data presented are not generally informative to distinguish the potential causal models, because of the noise? The performance of the analysis in the simulated data does not seem to be relevant unless the authors could artificially adjust the noise level to make the point of insufficient power.

-

- In most cases, the paper stops at the statistical analysis without attempting to explore the underlying biological mechanisms. I certainly don't expect the authors going after every identified CCV, but after reading the abstract, some positive examples of the shared biological mechanism for some strongly colocalized signals seem warranted.

- Just to be consistent with other analyses presented (e.g., eQTL mapping, interaction analysis), are the partial correlation analysis and mediation analysis controlled for multiple testing?

- I agree with the assessment that LD general complicates the resolution of colocalization analysis. In most cases, we can't identify the colocalized SNP but only the LD region. In light of this, we think the authors should be generally more careful in interpreting the subsequent analysis results based on the selected colocalized sites. This is particularly important in making statement of SNP-level effect size and SNP-level interaction patterns.

Reviewer #2 (Remarks to the Author):

The manuscript "Co-occurring eQTLs and mQTLs: detecting shared causal variants and shared biological mechanisms" by Pierce et al., is a comprehensive study that examines the extent to

which a shared causal variant underlies local genetic variation that affects gene expression and DNA methylation, as well as the causal relationship between expression and methylation. The authors study this in peripheral blood from a Bangladeshi population (992 with expression measurements and a separate set of 337 individuals with DNA methylation measurements for the colocalization analysis). The authors find that 48% of the lead variants for expression quantitative trait loci (eQTL) and methylation quantitative trait loci (mQTL) likely share a common causal variant (colocalize). More often than not, the direction of effect of the variant on expression and methylation is opposite, which is consistent with what is known about how methylation, for example in promoters, affects transcription. This study does a nice job in taking this finding one step further, by extensively investigating with three different methods, whether expression affects methylation and vice versa, for the eQTL-mQTL pairs that share a causal variant. It seems that larger sample sizes will be needed to draw more concrete conclusions on the causal relationship directionality between expression and methylation, as the authors discuss in the Discussion section and demonstrate with simulations and analyses (Supplementary Figure 3 and Supplementary Table 3). The authors do a nice job in discussing the limitations of the various methods used.

The interaction of eQTLs and mQTLs with age and sex feels a bit detached from the rest of the paper, and does not have the same level of depth and detail as the remainder of the paper.

This is a nice study that will be of interest to the scientific community, in particular people working on genetics of complex phenotypes, gene regulation and epigenetics. I have though several comments about some of the statistical analyses and significance estimations employed, which I think should be addressed to be able to draw solid conclusions about the shared regulation of expression and methylation, and the interplay between expression and methylation.

Major comments:

1. In the Methods section, under 'eQTL and mQTL analyses' on page 25, the authors describe the covariates included in the linear regression model. No genotype PCs were included, which could be Ok, but it would be useful to confirm this with PCA plots of the first few PCs, to see that there is no population stratification amongst the Bangladeshi individuals used in the study.
2. The approach used to correct for the multiple hypothesis burden in the cis-eQTL and cis-mQTL analyses (described on page 25, lines 475-476) - applying Benjamini Hochberg to all variant-gene/region pairs tested genome-wide - is a lenient approach and does not properly correct for differences in number of variants tested per gene or CpG region or for differences in linkage disequilibrium (LD) between all variants tested per gene/region. From experience, this will lead to a large inflation of results. An approach that accounts for these potential biases is described in GTEx (GTEx Science 2015) or on the GTEx portal: <http://www.gtexportal.org/home/documentationPage>, under the 'Analysis Methods' section, subheader 'eQTL Analysis'. Briefly, in this approach, q-values (Storey and Tibshirani, 2003) are computed on gene-level empirical p-values estimated with genotype-phenotype permutation analysis or beta distribution-extrapolation using the FastQTL tool. To identify a list of all significant variant-gene/region pairs associated with gene expression or DNA methylation, respectively, a nominal p-value threshold is computed for each gene, as described under 'Identification of all significant variant-gene pairs' in the Analysis Methods section on the portal. Applying this multiple hypothesis correction method, will likely lead to fewer significant eQTLs or mQTLs and may lead to a higher percentage of colocalizing eQTL and mQTL signals.
3. The authors use a colocalization method, coloc (Giambartolomei et al., PLoS Genetics, 2014). One of the limitations of this method is that it assumes that there is only one causal variant underlying the QTLs. This could lead to loss of detection power, and should at least be noted in the Methods and Results sections. New methods have been developed that consider the possibility of multiple causal variants underlying a QTL, such as eCAVIAR (Hormozdiari et al., AJHG 2016).
4. Is there a separate study (even from a different population background) that can be used to

test for replication of the colocalization results?

5. A $P < 0.05$ was used as the significance level for the partial correlation analysis and the mediation analysis (Pages 10 and 12, respectively). Is this after correcting for the number of pairs tested (e.g. using Bonferroni correction)?

6. In the partial correlation analysis, could the authors comment on whether the correlation that remains between expression and methylation after adjusting for the lead SNP, could be due to a secondary genetic association signal with expression and methylation.

7. Can the authors please describe how they computed SNP by sex or by age interactions? Also, what is the age range of the population studied, how well are the ages distributed, and what fraction of samples are females versus males? These factors could have an effect on the ability to detect genetic interaction with these variables.

Minor comments:

1. In the first paragraph in the introduction on page 3, where genome-wide scans of eQTLs in multiple human tissues is mentioned, a reference to the GTEx study (GTEx consortium, Science 2015 PMID: 25954001 and <http://biorxiv.org/content/early/2016/09/09/074450>) would be appropriate.

2. In the sentence on page 3, lines 63-65, I might add to the reasons why it is interesting to identify variants that have coordinated effects on multiple molecular phenotypes; see for example the words in bold:

"Because many QTLs appear to influence multiple local molecular phenotypes and since functional relationships exist between the different molecular phenotypes, there is great interest in identifying variants that have coordinated effects on multiple phenotypes and understanding the mechanisms by which such variants act."

3. In the last paragraph of the introduction on page 4, lines 83-84, it would be informative to mention from which cell types the expression and DNA methylation were measured.

4. On page 5, Figure 1, are the 9,629 unique lead eSNPs and 102,836 unique lead mSNPs in linkage equilibrium from each other for a given gene/region?

5. On page 5, lines 107-109, it would be informative to add a supplementary figure showing the distribution of distance of the lead eSNP to the transcript start site of the target gene and the lead mSNP to the affected CpG site.

6. In legend of Figure 3, gene symbols should be italicized.

7. In demonstrating a few examples of colocalizing cis-eQTL and cis-mQTL pairs (as shown in Figure 3), it would be informative to show a LocusZoom type of plot ($-\log P$ vs chromosomal coordinates) that colorcodes the mQTL points based on their level of LD (r^2) to the lead eQTL for the corresponding gene and vice versa.

8. On page 9, lines 165-167, the authors examine the association of the common causal variant probability with the local LD around the lead SNP (I assume the authors mean here 'eSNP?'). For a local LD measure, a sum of the pairwise r^2 between the lead SNP and SNPs in a 500kb window is used. I would recommend using an average of all pairwise r^2 values in the window instead of sum, as the density of variants may vary for different genes in the +/- 500kb window.

Second, I would recommend using a different name than "LD score", since it could be confused with the well known LD score regression method used to assess the relative contribution of functional regions in the genome to heritability of complex traits (Finucane et al, Nature Genetics 2015). A suggestion could be "LD measure".

9. On page 9, line 167 – was 'LD' meant to be 'lead'?

10. In Figure 5C, on page 11, is the x-axis '-log₁₀(P-value)'? Currently it states 'P-value', but it seems to me to be -logP.

11. In the Results section, when the numbers of significant pairs are reported for partial correlation, mediation and Bayesian network analyses, it would be useful to also report the percentage of total number of pairs tested in addition to the numbers.

12. On page 12, the authors investigate why the mediation analysis is not able to distinguish between the two causal relationship models (SNP->Expr->Methyl and SNP->Methyl->Expr). Based on the simulation results in Supplementary Figure 3, another explanation could be due to weak mediation effects.
13. On page 15, it would be useful to show a distribution of distance between the primary and secondary CpGs whose mQTL colocalizes with the same eSNP, for the 1219 cases mentioned.
14. Typo on page 22, line 413: 'and' should be 'any'
15. In the QC section of the genotyped variants on page 22 line 415, a HWE p-value cutoff of $<10E-10$ is used. Given that $\sim 299k$ variants were tested, I would recommend using a HWE cutoff of $p < 1.67E-07$ ($=0.05/299,000$).
16. On page 22, lines 416-417 – what version of the 1000 Genomes Project reference haplotypes was used for imputation?
17. On page 24, there is a discrepancy in the final number of individuals used for DNA methylation QTL analysis after QC (407, after removing individuals whose self reported sex did not match their sex based on methylation data) compared to the number noted on page 23 (400, after removing an additional 7 samples with $>5\%$ of CpGs with missing values or detection $p > 0.05$). Please clarify.
18. On page 24, line 454, was the intention to write $<30\%$ instead of $>30\%$?
19. On page 25, line 474, I think ' <500 Mb', was meant to be ' <500 kb'.
20. I have a terminology question – I have seen in the literature both meQTL and mQTL used for DNA methylation QTLs, and mQTL used for metabolite QTLs. meQTL seems to me more intuitive and more unique to methylation. I put it out there for the authors to consider what is used more often or is more accepted in the field of epigenetic regulation.
21. On page 26, line 509, where it is noted that the regression of the methylation probe and expression probe levels was done on the lead SNP, does this refer to the lead eSNP? Please specify.
22. On page 19, line 328, is 'expression' supposed to be 'methylation'?

Response to Reviewers' comments:

In responding to the reviewer comments, we have made several major changes to our analysis pipeline, including 1) a new method for FDR calculation for eQTL and meQTL analyses, 2) a new approaches for determining appropriate prior for Bayesian co-localization analyses, 3) replication of co-localization results using an independent dataset, and 4) multiple testing correction for partial correlation and mediation analyses. Point-by-point responses to the reviewers' comments are below. In addition, we have included a version of the manuscript with all changes marked at the end of the manuscript file.

Reviewer #1 (Remarks to the Author):

Pierce et al. discuss a joint expression QTL and methylation QTL analysis to identify potential common causal variants (CCVs) that impact both expression and methylation phenotypes. The analyses described in the manuscript are mostly carefully conducted, and the details are well documented. I do have a few concerns with respect to the conclusions drawn from the analyses, and I hope my comments are helpful for the authors to revise the manuscript.

Comment #1: The co-localization analysis needs to be improved. The authors are generally careful in the co-localization analysis using the coloc method. They clearly separate the non-overlapping samples for eQTL and mQTL analyses and perform the sensitivity analysis with respect to the different prior values. Nevertheless, in my opinion, there is a lack of justification why their final results rely on one set of prior (in particular $p_{12} = 1e-5$) vs. the others. Without controlling false positives, it is questionable to reject the more stringent threshold (10^{-7}) based on the fact that it would "eliminate vast majority of evidence for co-localization" (line 153, page 8). In general, I find that the data presented in this paper do not match the guideline given by the coloc method, largely because the coloc guideline is intended for GWAS where the signals are typically much more sparse. This discrepancy impacts not only the choice of prior parameter p_{12} but also key prior parameters p_1 and p_2 . Specifically based on the total number of interrogated SNPs, unique significant eQTL and mQTL SNPs in Table 1, I estimate p_1 (abundance of eQTLs) should be $\sim 10^{-3}$ (9629/8639940) and p_2 (abundance of mQTLs) should be $\sim 10^{-2}$. These are very different than the values used in the analysis (both are set at 10^{-4})

Response: In order to address the reviewer's concern regarding choice of p_1 and p_2 , we have calculated these priors based on our own data, as suggested. However, after carefully considering this issue, we have decided to take a slightly different approach than the one the reviewer suggests to obtain p_1 and p_2 . Using cis-meQTLs as an example, the probability that any given SNP is a causal variant affecting a nearby CpG site (ANY nearby CpG site) is $\sim 10^{-2}$, as pointed out by the reviewer (based on the fact we observe 64,483 unique lead meSNPs among 8,639,940 SNPs). However, for a co-localization test, the prior that should be used is the probability that a variant is causal for the nearby CpG that is being analyzed, NOT the probability that a SNP is causal for ANY nearby CpG site (as SNPs are often tested for association with many nearby CpG sites in a cis-mQTL analyses). So, to obtain p_1 , rather than divide the total number of mQTLs observed by the number of SNPs tested, our approach is to divide the number of "causal mQTL associations" observed (allowing SNPs to be associated with multiple CpGs, i.e., involved in multiple "causal mQTL association") by the total number of tests conducted. This will give us the approximate prior probability that a SNP is causal for a specific nearby CpG site (based on the 1Mb window we use in our analysis).

Based on our updated Table 1, we now observe that 77,664 CpG have an mQTL, represented by 64,483 lead SNPs (with some causal variants affecting multiple nearby CpGs). Under the simplifying assumption that each CpG has one primary causal variant (an assumption that used often

throughout this work), we have 77,664 unique SNP-CpG causal relationships. We have conducted 994,862,964 tests in our cis-mQTL analysis, which gives us a prior of $77,644/994,862,964 = 8 \times 10^{-5}$ (approximately 10^{-4}).

Similarly, for the eQTL analysis, we detected 6,788 gene probes with an eQTL, and we conducted 52,278,603 tests in our cis-mQTL analysis, which gives us a prior of $6,788/52,278,603 = 1.3 \times 10^{-4}$ (approximately 10^{-4}). Using the number of genes rather than probes gives us a prior of $5,632/52,278,603 = 1.1 \times 10^{-4}$. Thus, we feel it is reasonable to use the 10^{-4} for both p_1 and p_2 .

In order to address the reviewer's concern regarding our choice of p_{12} , we have adopted a method described by Guo et al. (Hum Mol Genet.2015 Jun 15;24(12):3305-13), a method they refer to as "internal empirical calibration". In summary, we consider a range of values for p_{12} , and select the value for which the posterior expectation of co-localization is similar to the prior expectation of co-localization. The results of this analysis are now presented as Figure 2. We tested p_{12} values of 5×10^{-5} , 1×10^{-6} , and 5×10^{-6} , corresponding to probabilities of an mQTL being an eQTL of 1/2, 1/10, and 1/100.

Because the prior is inversely related to the number of SNPs included in the co-localization analysis (as pointed out by Guo et al.), we re-analyzed our data using a smaller window (500 kb rather than 1 Mb). However, this approach gave us almost an identical number of co-localizing eQTL-mQTL pairs as we observed using a 1 Mb window.

We summarize the above reasoning in the Methods section, under the "Co-localization analyses" subsection.

Comment #2: The conclusion of the mediation (as well as partial correlation) analysis is very confusingly written (line 208 to 213). The purpose of the mediation analysis seems to be establishing the potential causal model between genetic variant, methylation level, and expression level. The authors should also make clear that pleiotropy is another potential causal model in consideration. After reading the text, my conclusion is that the data presented are not generally informative to distinguish the potential causal models, because of the noise? The performance of the analysis in the simulated data does not seem to be relevant unless the authors could artificially adjust the noise level to make the point of insufficient power.

Response: We have edited this section to improve the clarity as follows:

"In other words, evidence for mediation was often detected for specific gene-CpG pairs regardless of which causal model was tested (SEM or SME) (**Supplementary Figure 4A**). This demonstrates an important limitation of mediation analysis: while it can be useful for detecting evidence of a causal relationship, mediation analysis can be inadequate by itself for determining the direction of that causal relationship. However, we demonstrate using simulated data that evidence for mediation should be stronger when the causal model (SEM or SME) is correctly specified, even in the presence of measurement error (**Supplementary Figure 5**). In addition, statistical support for a correctly specified model will be more pronounced when the effects along the mediation pathway are stronger (**Supplementary Figure 5**). Lack of evidence of mediation implies either pleiotropy (i.e., no causal relationship between expression and methylation) or lack of power to detect mediation."

In other words, we agree that we cannot clearly distinguish between causal models. However, this is not due to noise or measurement error per se, it is a limitation of mediation analysis itself, which essentially is a test for shared variance among three variables. We have now included simulations that include random measurement error for both the CpG and the expression trait. These simulations demonstrate that, in the absence of measurement error, stronger evidence of mediation

will be detected if the correct model is specified. If the incorrect model is specified, evidence of mediation may still be detected, but the evidence will be weaker compared to analyses in which the correct model is specified.

Comment #3: In most cases, the paper stops at the statistical analysis without attempting to explore the underlying biological mechanisms. I certainly don't expect the authors going after every identified CCV, but after reading the abstract, some positive examples of the shared biological mechanism for some strongly co-localized signals seem warranted.

Response: We have now selected 3 examples where evidence for co-localization and mediation is strong. The examples we selected also have a relatively small number of potential causal SNPs (low LD score) and relevant to human traits. For these examples, we describe the association with expression and CpGs, target genes, overlap with genomic annotations, etc. These examples are described at the end of the Results section in the subsection entitled: *Examples of co-localized eQTL/meQTL pairs with strong evidence of mediation*

Comment #4: Just to be consistent with other analyses presented (e.g., eQTL mapping, interaction analysis), are the partial correlation analysis and mediation analysis controlled for multiple testing?

Response: We now use an FDR of 0.05 for the partial correlation analysis as well as both mediation tests (SME and SEM). This is based on 3,453 tests. We now report the numbers of significant tests for partial correlation analysis in the results section:

"Using an FDR of 0.05, we observed 233 eProbe-CpG pairs showing significant correlation after adjustment for the lead SNP, with 494 pairs showing partial correlation with $P < 0.05$ (Supplementary Figure 2)."

We also report the number of significant tests for mediation analysis in the results section: "Using an FDR of 0.05, we observed 168 eProbe-CpG pairs showing evidence of mediation under the SME model (Sobel $P < 0.0035$ and % mediation > 0), and 127 pairs showing evidence of mediation under the SEM model (Sobel $P < 0.0024$ and % mediation > 0). The two sets are largely overlapping, with 122 pairs showing mediation under both models, and 173 showing evidence of mediation under at least one model (Figure 5B)."

Comment #5: I agree with the assessment that LD general complicates the resolution of colocalization analysis. In most cases, we can't identify the colocalized SNP but only the LD region. In light of this, we think the authors should be generally more careful in interpreting the subsequent analysis results based on the selected colocalized sites. This is particularly important in making statement of SNP-level effect size and SNP-level interaction patterns.

Response: We agree that this is an important issue. In many places throughout the paper, we have changed the wording from "effect" to "association" when referring to results for a specific SNP (including the abstract, which now reads "These co-localized pairs are enriched for SNPs showing opposite associations with expression and methylation...").

We thank the reviewer for the thoughtful feedback and helping us improve the quality of this manuscript.

Reviewer #2 (Remarks to the Author):

The manuscript “Co-occurring eQTLs and mQTLs: detecting shared causal variants and shared biological mechanisms” by Pierce et al., is a comprehensive study that examines the extent to which a shared causal variant underlies local genetic variation that affects gene expression and DNA methylation, as well as the causal relationship between expression and methylation. The authors study this in peripheral blood from a Bangladeshi population (992 with expression measurements and a separate set of 337 individuals with DNA methylation measurements for the colocalization analysis). The authors find that 48% of the lead variants for expression quantitative trait loci (eQTL) and methylation quantitative trait loci (mQTL) likely share a common causal variant (colocalize). More often than not, the direction of effect of the variant on expression and methylation is opposite, which is consistent with what is known about how methylation, for example in promoters, affects transcription. This study does a nice job in taking this finding one step further, by extensively investigating with three different methods, whether expression affects methylation and vice versa, for the eQTL-mQTL pairs that share a causal variant. It seems that larger sample sizes will be needed to draw more concrete conclusions on the causal relationship directionality between expression and methylation, as the authors discuss in the Discussion section and demonstrate with simulations and analyses (Supplementary Figure 3 and Supplementary Table 3). The authors do a nice job in discussing the limitations of the various methods used.

The interaction of eQTLs and mQTLs with age and sex feels a bit detached from the rest of the paper, and does not have the same level of depth and detail as the remainder of the paper. This is a nice study that will be of interest to the scientific community, in particular people working on genetics of complex phenotypes, gene regulation and epigenetics. I have though several comments about some of the statistical analyses and significance estimations employed, which I think should be addressed to be able to draw solid conclusions about the shared regulation of expression and methylation, and the interplay between expression and methylation

Response: In response to the reviewer’s concern regarding the interaction analyses being detached from the rest of the paper, we have decided to remove these analyses from the paper.

Major comments:

Comment 1: In the Methods section, under ‘eQTL and mQTL analyses’ on page 25, the authors describe the covariates included in the linear regression model. No genotype PCs were included, which could be Ok, but it would be useful to confirm this with PCA plots of the first few PCs, to see that there is no population stratification amongst the Bangladeshi individuals used in the study.

Response: Our participants are very homogenous in terms of ancestry, showing no evidence of subgroups. Apologies for not making this clear. We have shown this in a prior paper (Pierce PLOS Genetics 2012). Applying PCA to unrelated individuals in our study produces no evidence of population structure. We have added the following text to the manuscript:

“No genotyping principle components were included because this is a very homogeneous cohort with no evidence of population strata (as previously reported (25, 26)), with eigenvalues from the first ten principle components being very similar (between 1.48 and 1.136). ”

Comment 2: The approach used to correct for the multiple hypothesis burden in the cis-eQTL and cis-mQTL analyses (described on page 25, lines 475-476) - applying Benjamini Hochberg to all

variant-gene/region pairs tested genome-wide - is a lenient approach and does not properly correct for differences in number of variants tested per gene or CpG region or for differences in linkage disequilibrium (LD) between all variants tested per gene/region. From experience, this will lead to a large inflation of results. An approach that accounts for these potential biases is described in GTEx (GTEx Science 2015) or on the GTEx portal: <http://www.gtexportal.org/home/documentationPage>, under the 'Analysis Methods' section, subheader 'eQTL Analysis'. Briefly, in this approach, q-values (Storey and Tibshirani, 2003) are computed on gene-level empirical p-values estimated with genotype-phenotype permutation analysis or beta distribution-extrapolation using the FastQTL tool. To identify a list of

all significant variant-gene/region pairs associated with gene expression or DNA methylation, respectively, a nominal p-value threshold is computed for each gene, as described under 'Identification of all significant variant-gene pairs' in the Analysis Methods section on the portal. Applying this multiple hypothesis correction method, will likely lead to fewer significant eQTLs or mQTLs and may lead to a higher percentage of colocalizing eQTL and mQTL signals.

Response: We thank the reviewer for this helpful suggestion. We have implemented the method as suggested. As the reviewer expected, we now detect fewer eQTLs and mQTLs (see the updated Table 1), as well as more frequent co-localization. The new text in the methods section ("eQTL and meQTL analyses" subsection) reads as follows:

"Linear regression implemented in the FastQTL software package³³ was used to conduct genome-wide *cis*-eQTL and *cis*-meQTL analyses. *Cis* associations were tested for SNPs and probes <500 kb apart using genotype dosages. For both the *cis*-eQTL and *meQTL* analyses, adaptive permutations were used in FastQTL (--permute 1000 10000) to obtain beta distribution-adjusted empirical p-values. A false-discovery rate (FDR) threshold of 0.01 was applied at the probe level (for both gene expression probe and CpG probes) using the *qvalue* package in R to identify probes with a significant QTL. SNP-probe pairs with a probe-level q-value <0.01 were defined as a significant eQTL-meQTL."

Comment #3: The authors use a colocalization method, *coloc* (Giambartolomei et al., PLoS Genetics, 2014). One of the limitations of this method is that it assumes that there is only one causal variant underlying the QTLs. This could lead to loss of detection power, and should at least be noted in the Methods and Results sections. New methods have been developed that consider the possibility of multiple causal variants underlying a QTL, such as *eCAVIAR* (Hormozdiari et al., AJHG 2016).

Response: We thank the reviewer for pointing out this issue. We now mention this limitation in the 5th paragraph of the discussion:

"First, the co-localization analysis approach we use estimates the probability of a single common causal variant and is not a test for multiple causal variants. Thus, it is possible that the presence of multiple causal variants or non-shared causal variants near a shared causal variant may reduce power to detect co-localization of a shared variant or variants. Recently developed methods can address this issue²⁷."

Comment #4: Is there a separate study (even from a different population background) that can be used to test for replication of the colocalization results?

Response: Yes! We recently generated a new batch of DNA methylation data (Illumina EPIC array) on an independent set of ~347 genotyped Bangladeshi individuals from a different study (HEALS, the Health Effects of Arsenic Longitudinal Study). We have conducted new *cis*-meQTL analyses for this data and conducted co-localization analyses using these new results and our existing eQTL results (restricting to CpGs present on both the EPIC and 450K arrays). Overall the co-localization

replication analyses were highly consistent with the primary analysis. We now present these results in Figure 3C.

The results section now includes the following: “We obtained meQTL results from an independent set of 347 unrelated Bangladesh individuals from the HEALS cohort (see methods) with DNA methylation data on ~850,000 CpG sites generated using the Illumina EPIC array. Using these cis-meQTL results and the eQTL results described above, we were able to attempt replication for 4,875 of the 5397 eQTL-meQTL pairs (522 pairs involved CpG sites measured on the 450K array but not on the EPIC array). Evidence of co-localization is consistent for the vast majority of the eQTL-meQTL pairs tested for replication (**Figure 3C**).

The methods section now includes a description of this new DNA methylation data (under “DNA methylation”) and meQTL analyses (under “eQTL and meQTL analyses”)

Comment #5: A $P < 0.05$ was used as the significance level for the partial correlation analysis and the mediation analysis (Pages 10 and 12, respectively). Is this after correcting for the number of pairs tested (e.g. using Bonferroni correction)?

Response: Reviewer 1 had a similar concern regarding multiple testing for partial correlation and mediation analyses. Our reply is below:

We now use an FDR of 0.05 for the partial correlation analysis as well as both mediation tests (SME and SEM). This is based on 3,453 tests. We now report the numbers of significant tests for partial correlation analysis in the results section:

“Using an FDR of 0.05, we observed 233 eProbe-CpG pairs showing significant correlation after adjustment for the lead SNP, with 494 pairs showing partial correlation with $P < 0.05$ (Supplementary Figure 2).”

We also report the number of significant tests for mediation analysis in the results section: “Using an FDR of 0.05, we observed 168 eProbe-CpG pairs showing evidence of mediation under the SME model (Sobel $P < 0.0035$ and % mediation > 0), and 127 pairs showing evidence of mediation under the SEM model (Sobel $P < 0.0024$ and % mediation > 0). The two sets are largely overlapping, with 122 pairs showing mediation under both models, and 173 showing evidence of mediation under at least one model (**Figure 5B**). ”

Comment #6: In the partial correlation analysis, could the authors comment on whether the correlation that remains between expression and methylation after adjusting for the lead SNP, could be due to a secondary genetic association signal with expression and methylation.

Response: This is an excellent point. We have added the following text to the results section:

“To explore the extent to which partial correlation could be due to secondary, co-localized causal variant affecting both the expression trait and the CpG being analyzed, we searched for secondary association signals for our 494 eQTL-meQTL pairs with partial correlation $P < 0.05$. We identified 92 pairs that had both a secondary eQTL and meQTL, and 35 of these pairs had a probability of $CCV > 0.80$. Among these pairs, 13 were no longer significant ($P < 0.05$) after adjusting for both the primary and secondary lead eSNP-meSNP (**Supplementary Table 3**), suggesting that a small fraction of our findings are affected by this issue.

Comment #7: Can the authors please describe how they computed SNP by sex or by age interactions? Also, what is the age range of the population studied, how well are the ages distributed, and what fraction of samples are females versus males? These factors could have an effect on the ability to detect genetic interaction with these variables.

Response: We have removed these analyses as suggested by Reviewer #1.

Minor comments:

1. In the first paragraph in the introduction on page 3, where genome-wide scans of eQTLs in multiple human tissues is mentioned, a reference to the GTEx study (GTEx consortium, Science 2015 PMID: 25954001 and <http://biorxiv.org/content/early/2016/09/09/074450>) would be appropriate.

Response: References added as suggested

2. In the sentence on page 3, lines 63-65, I might add to the reasons why it is interesting to identify variants that have coordinated effects on multiple molecular phenotypes; see for example the words in bold: “Because many QTLs appear to influence multiple local molecular phenotypes and since functional relationships exist between the different molecular phenotypes, there is great interest in identifying variants that have coordinated effects on multiple phenotypes and understanding the mechanisms by which such variants act.”

Response: Edited as suggested

3. In the last paragraph of the introduction on page 4, lines 83-84, it would be informative to mention from which cell types the expression and DNA methylation were measured.

Response: We now note the DNA and RNA are from peripheral blood.

4. On page 5, Figure 1, are the 9,629 unique lead eSNPs and 102,836 unique lead mSNPs in linkage equilibrium from each other for a given gene/region?

Response: We have pruned our list of lead eSNPs and lead meSNPs to obtain new lists that are free of any LD relationships with $r^2 > 0.5$. These numbers are now listed in the footnotes of Table 1:

“For the 6,526 unique lead eSNPs, after pruning out one SNP of each SNP pair with linkage equilibrium $r^2 > 0.5$, 5,385 independent SNPs remain. For the unique lead 64,483 meSNPs, 36,468 independent SNPs remain by using the same pruning method.”

5. On page 5, lines 107-109, it would be informative to add a supplementary figure showing the distribution of distance of the lead eSNP to the transcript start site of the target gene and the lead mSNP to the affected CpG site.

Response: We have added this as Supplementary Figure 6, which is now referenced in the line mentioned by the reviewer.

6. In legend of Figure 3, gene symbols should be italicized.

Response: edited as requested.

7. In demonstrating a few examples of colocalizing cis-eQTL and cis-mQTL pairs (as shown in Figure

3), it would be informative to show a LocusZoom type of plot (-logP vs chromosomal coordinates) that colorcodes the mQTL points based on their level of LD (r^2) to the lead eQTL for the corresponding gene and vice versa.

Response: We have now added a supplementary figure showing two locus zoom plots for each of the plots in figure 3, as requested by the reviewer (Supp Fig 3). We now reference this Figure in the text:

“These eQTLs and meQTLs signals are also shown, color-coded by LD with the lead meSNP and eSNP, respectively, in Supplementary Figure 3.”

8. On page 9, lines 165-167, the authors examine the association of the common causal variant probability with the local LD around the lead SNP (I assume the authors mean here ‘eSNP’?). For a local LD measure, a sum of the pairwise r^2 between the lead SNP and SNPs in a 500kb window is used. I would recommend using an average of all pairwise r^2 values in the window instead of sum, as the density of variants may vary for different genes in the +/- 500kb window.

Second, I would recommend using a different name than “LD score”, since it could be confused with the well known LD score regression method used to assess the relative contribution of functional regions in the genome to heritability of complex traits (Finucane et al, Nature Genetics 2015). A suggestion could be “LD measure”.

Response: We have changed “SNP” to “eSNP”. We are using the term “LD score” because we are calculating the LD score exactly as defined by Finucane et al (2015), which was initially described by Bulik-Sullivan et al. (2015). In their “LD score regression” method, they take a large set of SNPs and regress this score on the test statistics for each SNP. The LD score is intended to reflect both the number of SNPs in LD with an index SNP and the strength of those LD relationships. Both of these factors would be expected to affect the co-localization results. So by taking an average, one would actually lose the information on the number of SNPs in LD (perhaps reflecting SNP density) and be left with only the “average strength of LD per SNP”. We now clarify the rationale for this analysis in the Results section:

“It has been suggested that local LD patterns affect the posterior probability of CCV²⁰, so, using our co-localization results, we tested the association between the probability of a CCV for each probe-CpG pair and the “LD score” for each lead eSNP. The LD score was defined as the sum of the pairwise r^2 values between the lead eSNP and all SNPs within 500 kb²² (using LD data from unrelated Bangladeshi individuals) and represents the extent to which a SNP is correlated with nearby variants, capturing both strength of LD and quantity of correlated SNPs.”

9. On page 9, line 167 – was ‘LD’ meant to be ‘lead’?

Response: The reviewer is correct. We have made the change.

10. In Figure 5C, on page 11, is the x-axis ‘-log₁₀(P-value)’? Currently it states ‘P-value’, but it seems to me to be -logP.

Response: The reviewer is correct.. We have made this change.

11. In the Results section, when the numbers of significant pairs are reported for partial correlation, mediation and Bayesian network analyses, it would be useful to also report the percentage of total number of pairs tested in addition to the numbers.

Response: We now include the percentages in this suggestion, as suggested.

12. On page 12, the authors investigate why the mediation analysis is not able to distinguish between the two causal relationship models (SNP->Expr->Methyl and SNP->Methyl->Expr). Based on the simulation results in Supplementary Figure 3, another explanation could be due to weak mediation effects.

Response: This is an excellent point. With stronger effects along the mediation pathway would produce P-values and mediation estimates for SEM and SEM that are more divergent, providing stronger evidence for one causal model vs. another. We have added to following text to that section:

“In addition, statistical support for a correctly specified model will be more pronounced when the effects along the mediation pathway are stronger (**Supplementary Figure 5**).”

13. On page 15, it would be useful to show a distribution of distance between the primary and secondary CpGs whose mQTL colocalizes with the same eSNP, for the 1219 cases mentioned.

Response: We have added the following to the paper:

“The distribution of the distance between the primary and secondary CpGs overserved was <100 kb in ~75% of cases and is shown in Supplementary Figure 6”

14. Typo on page 22, line 413: ‘and’ should be ‘any’

Response: We deleted the “and”, which eliminates the typo.

15. In the QC section of the genotyped variants on page 22 line 415, a HWE p-value cutoff of <10E-10 is used. Given that ~299k variants were tested, I would recommend using a HWE cutoff of $p < 1.67E-07$ ($=0.05/299,000$).

Response: We initially elected to use a less stringent threshold because we performed QC on our entire genotyped samples (>5,000) which includes some relatives, leading to somewhat inflated HWE test statistics. We have now conducted HWE testing restricting to an unrelated set of participants, and we can confirm that no SNPs used in the analysis have an HWE P value < 10⁻⁷. We now state this in the methods section.

16. On page 22, lines 416-417 – what version of the 1000 Genomes Project reference haplotypes was used for imputation?

Response: The version is phase3 v5. We have now included this in the methods section.

17. On page 24, there is a discrepancy in the final number of individuals used for DNA methylation QTL analysis after QC (407, after removing individuals whose self reported sex did not match their sex based on methylation data) compared to the number noted on page 23 (400, after removing an additional 7 samples with >5% of CpGs with missing values or detection $p > 0.05$). Please clarify.

Response: Thank you for catching this inconsistency. We have deleted the last sentence of that paragraph. The confusions should now be resolved. The sample size after QC is 400.

18. On page 24, line 454, was the intention to write <30% instead of >30%?

Response: Correct. Thank you for catching this typo. It has been corrected.

19. On page 25, line 474, I think '<500 Mb', was meant to be '<500 kb'.

Response: Correct. Thank you for catching this typo. It has been corrected.

20. I have a terminology question – I have seen in the literature both meQTL and mQTL used for DNA methylation QTLs, and mQTL used for metabolite QTLs. meQTL seems to me more intuitive and more unique to methylation. I put it out there for the authors to consider what is used more often or is more accepted in the field of epigenetic regulation.

Response: Thank you for the helpful suggestion. We agree that meQTL appears to be more common in the literature. Our manuscript now uses “meQTL” throughout. In addition, we have replaced all occurrences of “mSNP” with meSNP”.

21. On page 26, line 509, where it is noted that the regression of the methylation probe and expression probe levels was done on the lead SNP, does this refer to the lead eSNP? Please specify.

Response: Yes. We use the lead eSNP in all partial correlation analyses and mediation analyses. This now reads “eSNP”

22. On page 19, line 328, is 'expression' supposed to be 'methylation'?

Response: you are correct. We have made this change.

We thank the reviewer for the thoughtful feedback and helping us improve the quality of this manuscript.

Reviewers' comments:

Reviewer #1 (Remarks to the Author):

I appreciate that the authors have thoughtfully responded my previous comments. I will be brief on the remaining technical issues.

1. I still have concerns on the prior choice of the coloc analysis. In particular, their presented approach to justify p_1 and p_2 makes little sense to me. Taken from the original coloc paper, p_1 and p_2 represent the (prior) probability that a SNP is associated with either of the two (molecular) traits. If those probabilities are added up across all SNPs (not gene-SNP pairs), the result provides a prior expectation of number of molecular QTLs. With this definition, it is very difficult for me to understand the justification to use all gene-SNP pairs as the denominator for estimating p_1 and p_2 , which does not seem to be consistent with the coloc model specification. At the same time, it is also very difficult to apply the authors' logic to define p_{12} .

2. The current prior has some implications that are probably too striking to comprehend. In particular, the authors assume 1 in 10^5 SNPs are being associated with expression, and at the same time, half of the mQTLs are eQTLs. The implied odds ratio of causal expression association between mQTL and baseline SNPs is just too large to believe, and it has a significant impact on the results.

The bottom line is that this particular colocalization analysis seems very sensitive to the prior specification, which generally is not a good sign for any Bayesian analysis. I am sympathetic in the sense that there is no standard way to get around this, however I am really uncomfortable to accept the current justification and result.

Reviewer #2 (Remarks to the Author):

The authors of the manuscript "Co-occurring eQTLs and mQTLs: detecting shared causal variants and shared biological mechanisms" have carefully and satisfactorily addressed the majority of my comments. I only have a few outstanding comments/edits, after which I think this manuscript is ready for publication.

1. In comment #6, I asked whether the correlation that remains between expression and methylation after adjusting for the lead SNP could be due to a secondary genetic association signal with expression and methylation. The authors have checked that and have added on page 7, line 173; "Among these pairs, 13 were no longer significant ($P < 0.05$) after adjusting for both the primary and secondary lead eSNP-meSNP (Supplementary Table 4), suggesting that a small fraction of our findings are affected by this issue."

I wouldn't portray this as 'an issue', but rather would present this as a result of potential interest - that a small fraction of pairs may be due to colocalization of secondary eSNP and meSNP signals (i.e. as something the people should consider, even if not the primary mechanism detected so far).

Also, there seems to be a typo: 'are adjusting' should be 'after adjusting'.

2. In the Mediation Analysis section on page 8, the authors find evidence for mediation under the SME (where DNA methylation mediates the effect of a SNP on local gene expression) and SEM model (where gene expression mediates the effect of a SNP on local DNA methylation). The

molecular mechanisms that may explain the effect of expression on DNA methylation are less understood (aside for changes in expression of DNA methyltransferases) than the effect of DNA methylation on expression (e.g. effect on transcription factor binding affinity). It would be interesting to check whether DNA methyltransferases are included amongst the target genes of eQTL/meQTL pairs that show mediation under the SEM model. To gain more biological insight from these results, it would also be informative to test whether the genes affected by eQTLs that mediate local DNA methylation are enriched in specific pathways or gene ontologies. This could also be interesting to do for target genes that fall under the SME model.

3. The authors chose to use meQTL as the abbreviation for DNA methylation QTLs. I just wanted to note that I still found a couple of 'mQTL' instead of 'meQTL' in the text.

Response to Reviewers' comments:

Reviewer #1 (Remarks to the Author):

I appreciate that the authors have thoughtfully responded my previous comments. I will be brief on the remaining technical issues.

1. I still have concerns on the prior choice of the coloc analysis. In particular, their presented approach to justify p_1 and p_2 makes little sense to me. Taken from the original coloc paper, p_1 and p_2 represent the (prior) probability that a SNP is associated with either of the two (molecular) traits. If those probabilities are added up across all SNPs (not gene-SNP pairs), the result provides a prior expectation of number of molecular QTLs. With this definition, it is very difficult for me to understand the justification to use all gene-SNP pairs as the denominator for estimating p_1 and p_2 , which does not seem to be consistent with the coloc model specification. At the same time, it is also very difficult to apply the authors' logic to define p_{12} .

Response: We agree with the reviewer that our choice of priors (p_1 and p_2) was not correct. We were incorrectly using the probability that a SNP is a cis-e/mQTL for a particular gene/CpG (rather than ANY gene/CpG) as the basis for p_1 and p_2 . We have now corrected this error, and we define p_1 and p_2 as follows (Methods section, "co-localization analysis" sub-section):

"For the eQTL analysis, we detected 5,022 independent eSNPs among 8,639,940 total SNPs, indicating the probability a SNP is a causal eSNP is 5.8×10^{-4} . This probability corresponds to the sum $p_1 + p_{12}$. For the meQTL analysis, we detected 29,472 independent meSNPs among 8,639,940 total SNPs, indicating the probability a SNP is a causal meSNP is 3.4×10^{-3} . This probability corresponds to the sum $p_2 + p_{12}$. Thus, our choice for p_{12} impacts the value of p_1 and p_2 . We varied the value of p_{12} (4.4×10^{-4} , 2.9×10^{-4} , and 1.45×10^{-4}) to correspond to probabilities of a causal eSNP being a causal meSNP of 75%, 50%, and 25%, which we view as a large and reasonable range for this prior."

Note that our approach is slightly different from prior studies in that we expect a substantial (and thus non-ignorable) proportion of eQTLs to also be mQTLs, so the value of p_1 and p_2 depend of the value selected for p_{12} . For example, the probability a SNP is a causal eSNP is $p_1 + p_{12}$, because p_1 corresponds to $P(\text{eQTL only})$ and p_{12} corresponds to $P(\text{eQTL and mQTL})$.

2. The current prior has some implications that are probably too striking to comprehend. In particular, the authors assume 1 in 10^5 SNPs are being associated with expression, and at the same time, half of the mQTLs are eQTLs. The implied odds ratio of causal expression association between mQTL and baseline SNPs is just too large to believe, and it has a significant impact on the results.

Response: This concern should be addressed in light of the updated values selected for p_1 and p_2 , in that p_1 is now a smaller probability than p_2 (i.e., the prior probability a SNP is a eSNP is smaller than the prior probability that a SNP is a meSNP).

The bottom line is that this particular colocalization analysis seems very sensitive to the prior specification, which generally is not a good sign for any Bayesian analysis. I am sympathetic in the sense that there is no standard way to get around this, however I am really uncomfortable to accept the current justification and result.

Response: Assuming the reviewer is referring to the sensitivity of the results to p_{12} , it is possible that the reviewer may have misinterpreted Figure 2 (the old Fig 2 is now Supplementary Fig 3). Our

co-localization results are represented by the dotted lines (with confidence bounds). The dotted lines do not change drastically as we vary the prior (p_{12}). The solid lines are the expected results assuming the priors are correct (not the actual result). It is well known that co-localization results are sensitive to the priors (as pointed out in prior papers), which makes selection of an appropriate range of priors very important (as well as post-analysis diagnostics). That being said, we do not feel our results are unusual in the context of co-localization analysis. In addition, we feel that the range of priors that we now use for p_{12} in the updated paper is broad and reasonable. We feel these are reasonable largely based on the fact that ~80% of our lead eSNPs are observed to be associated with DNA methylation (and we are measuring only a small fraction of CpGs in the human genome). As stated in the results section (3rd paragraph):

“5,192 of our 6,526 unique eSNPs were associated with methylation for at least one CpG among the 77,664 CpGs with a significant meQTL (FDR of 0.01), suggesting that a substantial number of causal eSNPs may also be causal meSNPs.”

In addition, there are biological reasons to expect substantial co-localization, as chromatin conformation is believed to both impact expression variation and be reactive to local transcriptional activity (SEM and SME).

To further address concerns regarding our choice of p_{12} , we now present our mediation and partial correlation analyses for “co-localized” eQTL-meQTL pairs for all three pre-specified values of p_{12} . As shown in supplementary figures (6-8 and 10-12), the patterns observed in these analyses are highly consistent regardless of the values selected for p_{12} (even though the numbers of pairs analyzed changes due to more “stringent” values selected for p_{12}).

In Summary, we feel the major messages of our paper are robust to this sensitivity of colocalization results to the selected prior (sensitivity which is expected when using the colocalization method).

Reviewer #2 (Remarks to the Author):

The authors of the manuscript “Co-occurring eQTLs and mQTLs: detecting shared causal variants and shared biological mechanisms” have carefully and satisfactorily addressed the majority of my comments. I only have a few outstanding comments/edits, after which I think this manuscript is ready for publication.

1. In comment #6, I asked whether the correlation that remains between expression and methylation after adjusting for the lead SNP could be due to a secondary genetic association signal with expression and methylation. The authors have checked that and have added on page 7, line 173; “Among these pairs, 13 were no longer significant ($P < 0.05$) after adjusting for both the primary and secondary lead eSNP-meSNP (Supplementary Table 4), suggesting that a small fraction of our findings are affected by this issue.”

I wouldn't portray this as ‘an issue’, but rather would present this as a result of potential interest - that a small fraction of pairs may be due to colocalization of secondary eSNP and meSNP signals (i.e. as something the people should consider, even if not the primary mechanism detected so far).

Also, there seems to be a typo: ‘are adjusting’ should be ‘after adjusting’.

Response: We thank the reviewer for pointing out this issue. The text now reads: “Among these pairs, 10 were no longer significant ($P < 0.05$) after adjusting for both the primary and secondary lead eSNP-meSNP (Supplementary Table 4), suggesting that a small fraction of our findings are due to co-localization of a secondary eQTL-meQTL pair.”

2. In the Mediation Analysis section on page 8, the authors find evidence for mediation under the SME (where DNA methylation mediates the effect of a SNP on local gene expression) and SEM model (where gene expression mediates the effect of a SNP on local DNA methylation). The molecular mechanisms that may explain the effect of expression on DNA methylation are less understood (aside for changes in expression of DNA methyltransferases) than the effect of DNA methylation on expression (e.g. effect on transcription factor binding affinity). It would be interesting to check whether DNA methyltransferases are included amongst the target genes of eQTL/meQTL pairs that show mediation under the SEM model. To gain more biological insight from these results, it would also be informative to test whether the genes affected by eQTLs that mediate local DNA methylation are enriched in specific pathways or gene ontologies. This could also be interesting to do for target genes that fall under the SME model.

Response: This is a good suggestion. However, no DNA methyltransferase were among the target genes of the eQTL/meQTL pairs (under SEM or SME, including DNMT1, DNMT3, and TRDMT1).

Regarding the mechanism underlying the SEM model, we now state in the Discussion (2nd paragraph):

“Prior studies have suggested that one of the mechanism underlying co-localized pairs is disruption of transcription factor (TF) binding sites, which can reduce TF binding affinity, thereby reducing transcriptional activity in the gene region and producing “reactive” changes in chromatin structure (including local DNA methylation) (16,20).”

In other words, the process of active transcription across the gene body (e.g., polymerases moving along the DNA) may contribute to chromatin remodeling and various epigenetic changes including DNA methylation. Thus, our original hypothesis regarding how the SEM model might work did not involve DNA methyltransferases, but it’s an interesting pathway to consider.

As suggested, we conducted gene set enrichment analyses for eGenes involved in the SEM and SME pathways. Using DAVID, we conducted analyses for SEM and SME genes using various GO classifications as well as KEGG, but none of the categories analyzed were significantly enriched in SEM or SME genes. We also conducted analyses using MSigDB, and two GO-MR categories were significantly enriched (FDR 0.01) for SEM genes: “cofactor binding” and “pyridoxal phosphate binding”. For SME we observed enrichment for “RNA binding”, “carbohydrate binding”, “polyA RNA binding”, “actin binding”, and “ubiquitin like protein transferase activity”. In our view, these results did not provide any specific information that helps us better interpret our findings, so our preference is not to include this in the manuscript.

3. The authors chose to use meQTL as the abbreviation for DNA methylation QTLs. I just wanted to note that I still found a couple of ‘mQTL’ instead of ‘meQTL’ in the text.

Response: Thank you for noticing this oversight. We have now removed replaced all instances of mQTL with meQTL.

Reviewers' comments:

Reviewer #1 (Remarks to the Author):

This revision addressed the issue raised in the previous review, especially for the choices of p_1 and p_2 priors. I find the current strategy and explanation more intuitive reasonable.

The only minor remaining issue is the choice of p_{12} . Although the empirical calibration analysis informed a choice of $p_{12} = 4.4 \times 10^{-4}$, the posterior colocalization results seemingly are more aligned with the choice of $p_{12} = 1.45 \times 10^{-4}$. (consider p_{12} is some sort of "empirical" prior, it should be close to the actual posterior overlapping fraction). This might be a flaw/problem of the empirical calibration analysis, but I think it is worth the attention of the authors. Additionally, there are now more published approaches available in the literature that can be used to estimate p_{12} , e.g.,

1. Pickrell, Joseph K., et al. "Detection and interpretation of shared genetic influences on 42 human traits." *Nature genetics* 48.7 (2016): 709-717.
2. Wen, Xiaoquan, et al. (2017). Integrating molecular QTL data into genome-wide genetic association analysis: Probabilistic assessment of enrichment and colocalization. *PLoS genetics*, 13(3), e1006646.
3. Giambartolomei, Claudia, et al. "A Bayesian Framework for Multiple Trait Colocalization from Summary Association Statistics." *bioRxiv* (2017): 155481.

These methods are obviously more sophisticated, I don't mean to ask the authors to re-do the analysis. But I hope that the authors should acknowledge that the choice of p_{12} has some significant impacts on determining the actual colocalized signals and discuss other alternatives to better estimate p_{12} .

Reviewer #2 (Remarks to the Author):

The authors of the manuscript "Co-occurring eQTLs and mQTLs: detecting shared causal variants and shared biological mechanisms" have carefully addressed our second round of comments. Based on the responses of the authors to both reviews, I have one main technical concern and a minor comment. If addressed, I think this paper will be of value to the genetic and functional genomics community.

1. The authors have fixed the way they estimate the prior probabilities that a SNP is a causal eSNP (p_1) or causal meSNP (p_2), which I think makes sense. If already the probabilities are conservative as they divide the number of independent eSNPs or meSNPs by the total number of SNPs tested, and not by the total number of independent SNPs tested.

My main comment is in regards to their choice of p_{12} , the prior probability that the eQTL and meQTL are tagging the same causal variant, based on the assumption that the probability of a causal eSNP being a causal meSNP (p_{12}) ranges between 25%-75%. The authors justify their choice based on the "internal empirical calibration" approach proposed by Guo et al., HMG 2015, that finds the p_{12} for which the posterior expectation of colocalization, averaged over all regions considered, most closely resembles the prior expectation of colocalization. I find this a bit circular. This is a good approach assuming the prior is correctly estimated.

From reviewing the literature it is not clear yet what fraction of eQTLs share a causal variant with meQTLs. It is not obvious that the proportion of eQTLs that share a causal variant with meQTLs is as high as 25%-75%. Variants affecting gene expression (eSNPs) may have other causal mechanisms, such as disruption of splice sites or transcription factor binding motifs. It has been shown that meQTL have a high predictive power for an eQTL but the predictive power in the other direction is less clear. The authors state in the manuscript: "5,192 of our 6,526 unique eSNPs were associated with methylation for at least one CpG among the 77,664 CpGs with a significant meQTL (FDR of 0.01), suggesting that a substantial number of causal eSNPs may also be causal meSNPs", but I would claim that it is hard to predict the p_{12} probability from the observed overlap.

Since colocalization analysis is sensitive to the choice of prior probabilities, I think it would be

prudent to present colocalization results for lower p_{12} values, such as based on the assumption that only 5%-10% of the causal eSNPs are also causal meSNPs.

2. The authors found that no DNA methyltransferases were among the target genes of the eQTL/meQTL pairs under the SEM or SME models. I would recommend mentioning this in the Results. I think it could be informative to include in the GSEA results in the supplementary material, but I respect the authors' choice not to include the results, as it is not critical to the interpretation of their results.

Response to Reviewers' comments:

Reviewer #1 (Remarks to the Author):

This revision addressed the issue raised in the previous review, especially for the choices of p_1 and p_2 priors. I find the current strategy and explanation more intuitive reasonable.

The only minor remaining issue is the choice of p_{12} . Although the empirical calibration analysis informed a choice of $p_{12} = 4.4 \times 10^{-4}$, the posterior colocalization results seemingly are more aligned with the choice of $p_{12} = 1.45 \times 10^{-4}$. (consider p_{12} is some sort of "empirical" prior, it should be close to the actual posterior overlapping fraction). This might be a flaw/problem of the empirical calibration analysis, but I think it is worth the attention of the authors. Additionally, there are now more published approaches available in the literature that can be used to estimate p_{12} , e.g.,

1. Pickrell, Joseph K., et al. "Detection and interpretation of shared genetic influences on 42 human traits." *Nature genetics* 48.7 (2016): 709-717.
2. Wen, Xiaoquan, et al. (2017). Integrating molecular QTL data into genome-wide genetic association analysis: Probabilistic assessment of enrichment and colocalization. *PLoS genetics*, 13(3), e1006646.
3. Giambartolomei, Claudia, et al. "A Bayesian Framework for Multiple Trait Colocalization from Summary Association Statistics." *bioRxiv* (2017): 155481.

These methods are obviously more sophisticated, I don't mean to ask the authors to re-do the analysis. But I hope that the authors should acknowledge that the choice of p_{12} has some significant impacts on determining the actual colocalized signals and discuss other alternatives to better estimate p_{12} .

Response: We appreciate these helpful suggestions, and we have made various changes to the text in response. First, we now highlight the impact of the choice of priors on the co-localization results in the results section, 4th paragraph (note that we now use 5 values for p_{12} , rather than 3, based on the comments from review 2):

"However, the number of pairs passing this threshold depended strongly on the value of the prior p_{12} , ranging from 2,913 such pairs when p_{12} was set of 4.4×10^{-4} to 266 pairs when p_{12} was set to 2.9×10^{-5} (**Table 2**). Due to uncertainty regarding the appropriate value for this prior, we conduct downstream analyses of "co-localized" pairs for each of the five values used for p_{12} ."

As mentioned in this new text, we have also included a new table (Table 2) that describes the priors used and the number of "co-localized" eQTL-meQTL pairs observed for each prior, which clearly shows the sensitivity of the co-localization results to the prior.

In order to address recent developments in prior estimation and limitations our approach (including the "empirical prior" issue pointed out by the reviewer), we have now included the following text in the discussion section (end of 3rd paragraph):

"Specifying appropriate priors for co-localization is also a challenge, considering posterior probabilities are sensitive to the choice of priors^{22,28}. However, there are several recently-proposed

methods that use genome-wide summary statistics for both traits and analyses of enrichment to estimate these priors²⁹⁻³¹, thereby avoiding subjective decisions regarding prior specification. While the “internal empirical calibration” approach we used suggested that 4.4×10^{-4} was the best choice for p_{12} , the number of instances of co-localization detected was consistently smaller than the prior expectation for “true” co-localized pairs for all five values of p_{12} , with smaller discrepancies observed for smaller values for p_{12} (**Table 2**). These discrepancies may be due in part to limited statistical power for co-localization analysis of weak QTL signals or may reflect a limitation of the internal empirical calibration approach.”

Reviewer #2 (Remarks to the Author):

The authors of the manuscript “Co-occurring eQTLs and mQTLs: detecting shared causal variants and shared biological mechanisms” have carefully addressed our second round of comments. Based on the responses of the authors to both reviews, I have one main technical concern and a minor comment. If addressed, I think this paper will be of value to the genetic and functional genomics community.

Comment 1: The authors have fixed the way they estimate the prior probabilities that a SNP is a causal eSNP (p_1) or causal meSNP (p_2), which I think makes sense. If already the probabilities are conservative as they divide the number of independent eSNPs or meSNPs by the total number of SNPs tested, and not by the total number of independent SNPs tested.

My main comment is in regards to their choice of p_{12} , the prior probability that the eQTL and meQTL are tagging the same causal variant, based on the assumption that the probability of a causal eSNP being a causal meSNP (p_{12}) ranges between 25%-75%. The authors justify their choice based on the “internal empirical calibration” approach proposed by Guo et al., HMG 2015, that finds the p_{12} for which the posterior expectation of colocalization, averaged over all regions considered, most closely resembles the prior expectation of colocalization. I find this a bit circular. This is a good approach assuming the prior is correctly estimated.

From reviewing the literature it is not clear yet what fraction of eQTLs share a causal variant with meQTLs. It is not obvious that the proportion of eQTLs that share a causal variant with meQTLs is as high as 25%-75%. Variants affecting gene expression (eSNPs) may have other causal mechanisms, such as disruption of splice sites or transcription factor binding motifs. It has been shown that meQTL have a high predictive power for an eQTL but the predictive power in the other direction is less clear. The authors state in the manuscript: “5,192 of our 6,526 unique eSNPs were associated with methylation for at least one CpG among the 77,664 CpGs with a significant meQTL (FDR of 0.01), suggesting that a substantial number of causal eSNPs may also be causal meSNPs”, but I would claim that it is hard to predict the p_{12} probability from the observed overlap.

Since colocalization analysis is sensitive to the choice of prior probabilities, I think it would be prudent to present colocalization results for lower p_{12} values, such as based on the assumption that only 5%-10% of the causal eSNPs are also causal meSNPs.

Response: We appreciate the reviewer’s perspective on this issue, and we agree using a wider range of priors will improve this work. We now report results for co-localization analyses using two additional priors for p_{12} , specially 5.8×10^{-5} and 2.9×10^{-5} (corresponding to probabilities of 10% and 5% that a causal eSNP is a causal meSNP, respectively). These additional analyses are described in the “Co-localization of cis-eQTLs and cis-meQTLs” section of the Results section:

“We varied the value of p_{12} (4.4×10^{-4} , 2.9×10^{-4} , 1.45×10^{-4} , 5.8×10^{-5} , and 2.9×10^{-5}) to correspond to probabilities of a causal eSNP being a causal meSNP of 75%, 50%, 25%, 10%, and 5% respectively, which we view as a large and reasonable range for this prior.”

We have now updated Figures 2 and 4 to reflect this change. Also, we’ve added several additional supplementary figures to show mediation and partial correlation results that are based on these new co-localization results (Supp Figs 8, 9, 15, 16). We have also updated several other figures with changes that reflect these additional analyses (Supp Figs 3, 10, and 12). Our general conclusion is that while the number of co-localized pairs depends strongly on the co-localization priors, the patterns we observe for mediation and partial correlation analysis are fairly consistent regardless of what priors are used for co-localization analysis. We state this in the “*Comparison of Partial Correlation and Mediation Results*” section of the results:

“The proportion of co-localized pairs showing evidence of mediation and/or partial correlation ($P < 0.05$) was 16%, 18%, 24%, 19%, and 17% for the priors 2.9×10^{-4} , 1.45×10^{-4} , 5.8×10^{-5} , and 2.9×10^{-5} , respectively.”

We also mention this point in the discussion section (first paragraph):

“The proportion of co-localized pairs showing evidence of mediation (and/or partial correlation) was fairly consistent regardless of the prior used, varying between 15%-24%.”

Comment 2: The authors found that no DNA methyltransferases were among the target genes of the eQTL/meQTL pairs under the SEM or SME models. I would recommend mentioning this in the Results. I think it could be informative to include in the GSEA results in the supplementary material, but I respect the authors’ choice not to include the results, as it is not critical to the interpretation of their results.

Response: As suggested, we now mention this finding in the paper. However, it seemed to fit best in the discussion section, 2nd paragraph, in which we describe potential mechanisms that could produce co-localized eQTL-meQTL pairs:

“Interestingly, there were no DNA methyltransferases (e.g., *DNMT1*, *DNMT3*, and *TRDMT1*) among the gene pairs classified as SEM (or SME).”

We appreciate the reviewer’s flexibility regarding not including the GSEA results. It’s really a lot of material that unfortunately does not appear to provide an important insight into our results.

REVIEWERS' COMMENTS:

Reviewer #2 (Remarks to the Author):

The authors have addressed my final concern and I think the paper is ready for publication. They have diligently tested the effect of the choice of p_{1_2} (the prior probability that an eQTL and meQTL are tagging the same causal variant) on the posterior probability that two co-occurring eQTL and meQTL signals share the same causal variant, by testing a range of p_{1_2} . This also discuss this issue in the discussion.